# RAIN: machine learning-based identification for HIV-1 bNAbs

Mathilde Foglierini [1,2,3,11], Pauline Nortier[1,2,11], Rachel Schelling[1,2], Rahel R. Winiger [1,2], Philippe Jacquet [4], Sijy O'Dell[5], Davide Demurtas[6], Maxmillian Mpina[7], Omar Lweno [7], Yannick D. Muller [1,2], Constantinos Petrovas[8], Claudia Daubenberger [9,10], Matthieu Perreau[1], Nicole A. Doria-Rose [5], Raphael Gottardo [3] & Laurent Perez [1,2] ✉

Broadly neutralizing antibodies (bNAbs) are promising candidates for the treatment and prevention of HIV-1 infections. Despite their critical importance, automatic detection of HIV-1 bNAbs from immune repertoires is still lacking. Here, we develop a straightforward computational method for the Rapid Automatic Identification of bNAbs (RAIN) based on machine learning methods. In contrast to other approaches, which use one-hot encoding amino acid sequences or structural alignment for prediction, RAIN uses a combination of selected sequence-based features for the accurate prediction of HIV-1 bNAbs. We demonstrate the performance of our approach on non-biased, experimentally obtained and sequenced BCR repertoires from HIV-1 immune donors. RAIN processing leads to the successful identification of distinct HIV-1 bNAbs targeting the CD4-binding site of the envelope glycoprotein. In addition, we validate the identified bNAbs using an in vitro neutralization assay and we solve the structure of one of them in complex with the soluble native-like heterotrimeric envelope glycoprotein by single-particle cryo-electron microscopy (cryo-EM). Overall, we propose a method to facilitate and accelerate HIV-1 bNAbs discovery from non-selected immune repertoires.

More than 40 years after its identification, the human immunodeficiency virus-1 (HIV-1) remains a major global health concern[1]. The World Health Organization (WHO) estimates 38 million HIV-1 infected individuals worldwide in 2023, 1.5 million new HIV-1 infections, and 650,000 deaths from acquired immunodeficiency syndrome (AIDS)-related illness. Despite intense research efforts, there is still no cure nor vaccine for HIV-1 infections available[2]. Humoral immune response to HIV-1 targets the envelope (Env) protein of the virion, a trimeric membrane glycoprotein complex comprising gp120 and gp41[3]. However, the virus rapidly escapes immune control due to the exceptional Env glycoprotein diversity generated by the error-prone replication machinery of HIV-1[4]. Moreover, additional mechanisms of immune evasion exist, such as heavy glycosylation of gp120, which promotes a conformational masking of the receptor-binding site[5]. Screening of

[1]Department of Medicine, Service of Immunology and Allergy, Lausanne University Hospital and University of Lausanne, Lausanne, Switzerland. [2]Centre for Human Immunology, Lausanne, Switzerland. [3]Biomedical Data Science Centre, Lausanne University Hospital and University of Lausanne, Lausanne, Switzerland. [4]Scientific Computing and Research Support Unit, University of Lausanne, Lausanne, Switzerland. [5]Vaccine Research Center, National Institute of Allergy and Infectious Diseases, National Institutes of Health, Bethesda, MD, USA. [6]Interdisciplinary center of electron microscopy, CIME, Ecole Polytechnique Fédérale de Lausanne, Lausanne, Switzerland. [7]Ifakara Health Institute, Bagamoyo, United Republic of Tanzania. [8]Department of Laboratory Medicine and Pathology, Institute of Pathology, Lausanne University Hospital, Lausanne, Switzerland. [9]Department of Medical Parasitology and Infection Biology, Clinical Immunology Unit, Swiss Tropical and Public Health Institute, Basel, Switzerland. [10]University of Basel, Basel, Switzerland. [11]These authors contributed equally: Mathilde Foglierini, Pauline Nortier. ✉e-mail: laurent.perez@chuv.ch

plasma from HIV-1 seropositive (HIV-1 + ) subjects led to the identification of rare individuals possessing sera with broad and potent neutralizing activities against numerous HIV-1 viruses. Additional studies allowed the cloning and sequencing of B-cell receptors (BCRs) and permitted the identification of broadly neutralizing antibodies (bNAbs), which can neutralize most viral strains at low concentrations in vitro[6]. Investigation of the development and structural properties of these bNAbs revealed only a low level of sequence identity between them, but demonstrated that specific characteristics are associated with their function. For example, bNAbs exhibit an extreme level of somatic hypermutations (SHMs) and large nucleotide insertions leading to long heavy chain complementary determining regions (CDRs)[7,8].

Since their identification, bNAbs have gained intense therapeutic interest. Although approved drugs against HIV-1 infection exist, passive antibody prophylaxis and immunotherapy could hold a valuable place in both prevention and treatment[9]. Passive transfer of bNAbs demonstrated a decrease of viral loads[10,11], prevention of infection[12,13], delay of viral rebound[14,15], and suppression of viremia in humanized mice, non-human primates, and humans without notable adverse events or side effects[16,17]. BNAbs target distinct sites of vulnerability at the surface of the envelope: the CD4-binding site (CD4bs), variable loop V1/V2 apex, and V3 loop, a larger site spanning the interface between gp41 and gp120 (interface) including the fusion peptide, and the membrane-proximal external region (MPER). Recently, a sixth site was discovered, defined by the bNAb VRC-PG05, which binds to the center of the so-called "silent face" of gp120[18].

To date, the identification of bNAbs has required B-cell isolation and clonal expansion from selected individuals possessing sera with broadly neutralizing activity. This step is followed by antibody cloning and experimental validation of their neutralization potential. While both steps represent an important research effort, the process has benefited from identified immune donors[19] and the development of high-throughput analyses of antibody repertoires by next-generation sequencing (NGS). Still, the number of identified HIV bNAbs remains relatively low, with only 255 of them being reported[3,20]. Some bNAbs have been investigated in registered clinical trials, for prevention, as a component of long-acting antiretroviral therapy (ART), or intervention aimed at long-term drug-free remission of HIV[17,21,22]. However, the clinical success of bNAb passive immunization strategies will likely require a combination of antibodies to increase the overall breadth and potency against diverse HIV-1 isolates and to prevent the emergence of resistance[23]. The recent deployment of large datasets of human B-cell repertoires on database repositories represents an opportunity for novel bNAb identification assuming that computational tools for their automatic identification and classification are developed[24]. Artificial intelligence (AI)-based prediction tools to find the antibodies and antigens have been developed[25]. However, most of these tools rely on structural or amino acid sequence similarities of related antibodies to identify potential target proteins[26]. Nonetheless, despite important research and characterization efforts, a precise set of criteria required for classifying bNAbs versus non-bNAbs is still lacking.

Here, we developed a computational pipeline named RAIN for the Rapid Automatic Identification of bNAbs from Immune Repertoires. RAIN is based on four different machine-learning algorithms, which can be trained in just a few minutes using a Python script. RAIN only requires the following: a cellranger scBCR output going through the Immcantation pipeline, and a R script converting the repertoire data into a features table for bNAb prediction. We validated RAIN on previously identified bNAbs, leading to a prediction accuracy of 100% and an Area Under the Curve (AUC) value ranging from 0.92 to 1, depending on the antigenic site. In addition, we isolated class-switched memory B cells from HIV-1 immune donors and performed single-cell BCR sequencing to demonstrate the method's performance. Importantly, immune repertoire analysis of donors with a serum able to broadly neutralize different HIV-1 isolates led to the identification of three bNAbs, while none was detected in the repertoire of immune donors with sera that did not possess broadly neutralizing activities. The identified bNAbs were further assessed for their affinities to the stabilized envelope prefusion trimer BG505 DS-SOSIP, for their neutralizing activities and one of them was additionally characterized by cryo-EM.

## Results

### Subset of discrete characteristics discriminates HIV-1 bNAbs from mAbs

The automatic identification of HIV bNAbs cannot be solely based on amino acid sequence similarity of the heavy or light chains, due to a large sequence variability resulting from the long affinity maturation process. In contrast, HIV-1 bNAbs isolated from chronically infected adults exhibit a signature of characteristic features, including high SHMs, insertions or deletions (indels), long complementarity-determining regions H3 (CDRH3), high potency, and broad viral neutralization breadth[3]. Moreover, the VRC01-class bNAbs, targeting the CD4bs, have also been shown to preferentially use specific germline alleles[27,28] and possess an unusually short CDRL3 of only five amino acids. These short CDRL3 are essential to contact gp120, while avoiding the glycan at position N267 in the D loop of gp120[29]. While bNAbs targeting the V1V2 apex use specific IGHV genes and together with bNAbs binding the V3 glycan, they are characterized by a long (20–34 residues) CDRH3 sequence[30,31]. We hypothesized that integrating specific parameters characterizing HIV-1 bNAbs in a machine-learning framework could allow a rapid identification of bNAbs from an immune repertoire (Fig. 1). To identify predictors of HIV-1 bNAbs, we investigated specific features associated with these antibodies and inferred them from their highly diversified amino acid sequences. We collected and curated bNAb sequences from the CATNAP (Compile, Analyze, and Tally NAb Panels) database[32]. Data curation consisted of only considering human affinity matured sequences and removing incomplete or unpaired sequences (Supplementary Data file 1). We obtained a total of 255 bNAb paired sequences, described to bind the V1V2 apex ($n = 98$), V3 glycan ($n = 56$), CD4-binding site ($n = 54$), gp120/gp41 interface ($n = 26$), and MPER ($n = 21$). To visualize the sequence similarity among these selected bNAbs, we initially aligned the sequences using ANARCI with the IMGT format[33]. Subsequently, we computed an identity score matrix to represent the inverse of the Levenshtein distance for each sequence pair. We selected either the full-length variable heavy (VH) or only the CDRH3 amino acid sequence (Fig. 2a). As expected, VH and CDRH3 share only minimal conservation among HIV-1 bNAbs. This result indicates that a homology and alignment approach to identify bNAbs would probably be unsuccessful. Next, to create a dataset of paired BCR sequences that is unlikely to recognize an HIV antigen (hereafter named mAbs), we retrieved and curated paired antibody sequences from ten healthy seronegative donors to obtain a total of 14,962 sequences (Supplementary Table 1). To control the comparability of the HIV-1 bNAbs with unassigned mAbs, we performed an additional similarity matrix with VH and CDRH3 comparing bNAbs and curated mAbs (Fig. 2b). As anticipated, only low similarity levels were observed, a result in agreement with the precedent matrix and indicating that the sequences were amenable to machine-learning approaches.

We then decided to investigate if some of the bNAbs distinct properties could be used as predictive variables for each targeted antigenic site. We considered as potential predictors the length of the CDR3 for the heavy (H3) and light (L3) chains, the frequency of SHM in the V gene (v) or unconventional acquired mutations in the framework regions only (uv), and the hydrophobicity of CDRH3[34,35] ($\varphi$) (Fig. 3a–e). Interestingly, anti-CD4bs bNAbs analysis demonstrated a statistically higher SHM frequency, a higher frequency of unconventional mutations (outside of the CDRs)[35], and a significantly shorter length of CDRL3 (Fig. 3a, b, e and Supplementary Fig. 1a) compared to the control mAbs reported in Supplementary Table 1. For the anti-MPER

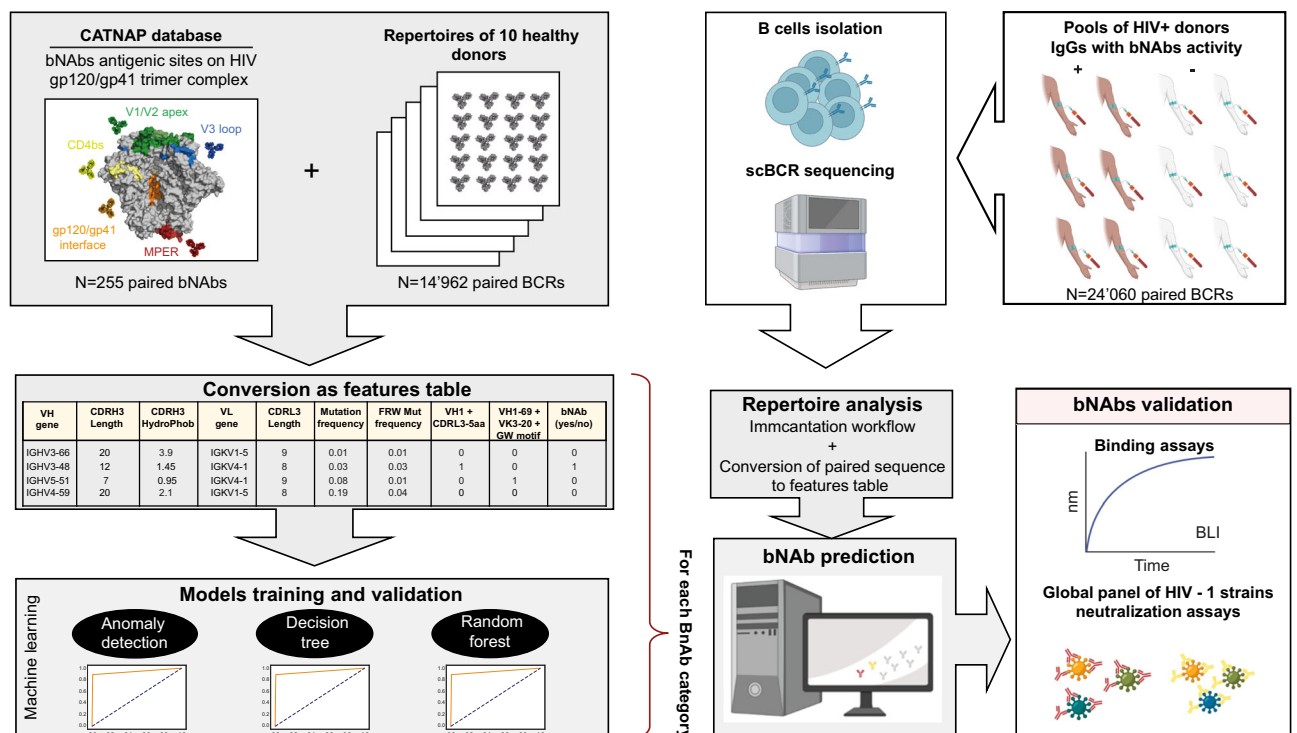

**Fig. 1 | RAIN pipeline for automatic identification of bNAbs.** Data collected from the CATNAP database (bNAbs) and healthy donor repertoires (mAbs) were converted into a features table to train and validate four machine-learning models: anomaly detection (AD), Decision Tree (DT), random forest (RF), and super learner (SL). We performed single-cell BCR sequencing from HIV-1 seropositive donors with

sera exhibiting broadly neutralizing activities (illustrated by the brown arms) or without neutralizing activities (illustrated by the white arms). BCR sequences were processed by the Immcantation workflow and analyzed as a feature table. Next, the predicted bNAbs found by the four algorithms were produced and tested in neutralization and binding assays. The image was created using BioRender.com.

bNAbs, we observed a longer CDRH3, with higher hydrophobicity, and a higher mutation frequency in both V gene and framework (FRW) regions (Fig. 3a–d and Supplementary Fig. 1b). The bNAbs targeting the V1V2 apex showed a higher mutation frequency of the V gene, but the difference was mainly due to a higher hydrophobicity of the CDRH3 and a longer CDRH3 (Fig. 3a–d and Supplementary Fig. 1c). BNAbs targeting the V3 glycan have a higher mutation frequency, slightly higher hydrophobicity of the CDRH3 and a longer CDRH3 (Fig. 3a–d and Supplementary Fig. 1d). BNAbs targeting the interface region demonstrated an increased frequency of mutations in the V gene and FWR regions (Fig. 3a, b and Supplementary Fig. 1e). Part of these results were expected but confirmed that this set of characteristics is statistically different between bNAbs and mAbs. To further investigate if these characteristics could be used to discriminate between bNAbs and mAbs, we decided to use them as variables in a two-dimensional Principal Component Analysis (PCA) (Fig. 3f–j). Remarkably, the five characteristics were sufficient to separate bNAbs from mAbs into two distinct clusters within each category of antigenic sites. We observed an explained variation of 0.43 for PC1 and 0.29 for PC2 across all five antigenic sites, while the weights of the features exhibited striking similarities. For PC1, the frequency of mutation in both, the CDRs and framework regions were important, whereas the hydrophobicity and length of CDRH3 were important for PC2. Unexpectedly, the length of CDRL3 was a less important feature. Based on these observations, we decided to use this set of measurable characteristics as predictors to distinguish bNAbs from mAbs.

**Algorithm selection and validation for the computational pipeline**

To further investigate the feasibility of automatic identification of potential HIV-1 bNAbs, we decided to use different machine-learning approaches to increase robustness and decrease the likelihood of false

predictions. First, antibody sequences were converted into a list of values corresponding to the set of predictors identified previously. BNAb sequences coming from the CATNAP database were annotated using Igblast and the Immcantation workflow[36–38]. The resulting Adaptive Immune Receptor Repertoire (AIRR) characteristics were converted to a feature format table. Similarly, mAb sequences obtained from public databases were processed as described previously[39] and converted into a features table. For each antigenic site, bNAbs and mAbs were pooled as one dataset and subdivided into three: 60% as a training set and 20% each as a validation and test set, respectively. An anomaly detection (AD) algorithm has been used in the specific case of a binary classification task, where one group appears as an outlier[40]. Given the scarcity of reported HIV-1 bNAbs compared to the quantity of mAbs, we first opted for the AD algorithm to automatically identify bNAbs. We used the multivariate Gaussian model based on a threshold value (Epsilon) to estimate the probability of an antibody being flagged as 'anomaly' or not. Then, the optimal Epsilon parameter minimizing the number of false positives was obtained using the validation set (Supplementary Fig. 2a–e and Supplementary Table 2), while the evaluation of the AD performance, including computing of the area under the curve (AUC) was done with the test set (Fig. 4a, b). We observed that the AD algorithm discriminates well bNAbs targeting the V1V2 apex (AUC: 0.93), the CD4bs (AUC: 0.88), the MPER (AUC: 0.82), and the interface (AUC: 0.8). However, bNAbs targeting the V3 glycan were poorly identified, with an AUC of 0.64. Moreover, a high number of false positives was obtained, indicating a low precision with the AD (Fig. 4a). To increase recall and precision of our detection method, we used both Decision Tree (DT) and random forest (RF) algorithms.

First, we used a random forest to analyze the identification profile of bNAbs with two classifying features and found that it allowed a clear decision boundary plot on the training dataset for bNAbs targeting the

**a**

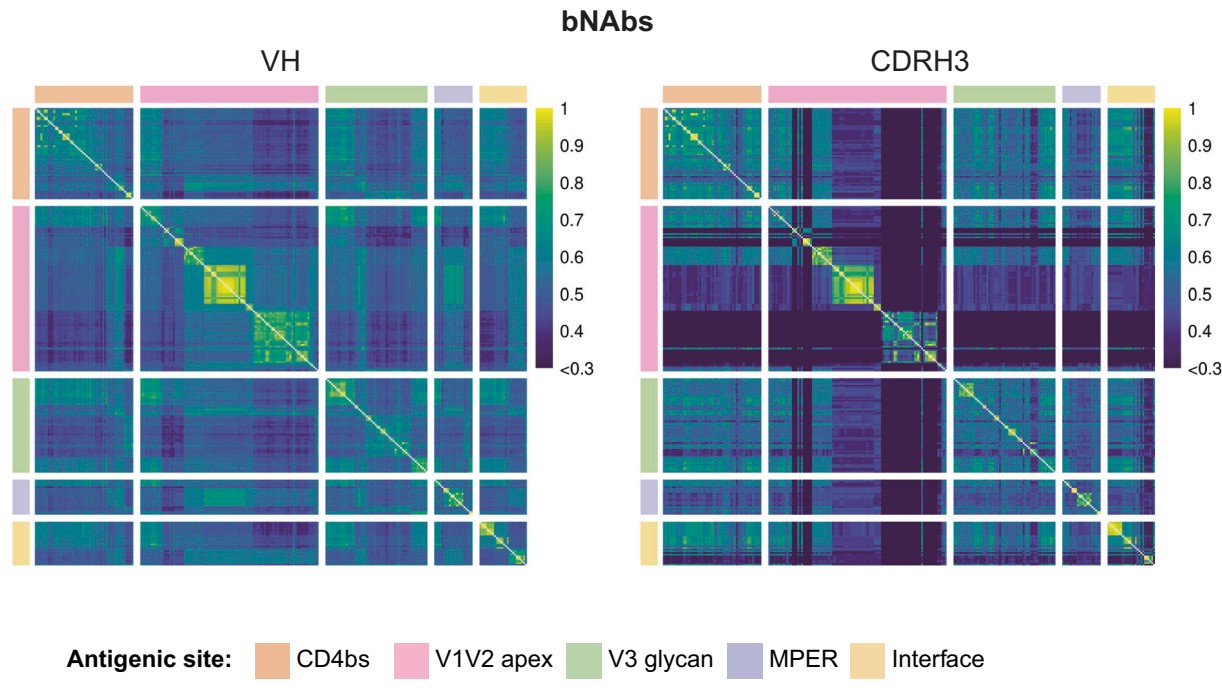

**Antigenic site:** ▮ CD4bs   ▮ V1V2 apex   ▮ V3 glycan   ▮ MPER   ▮ Interface

**b**

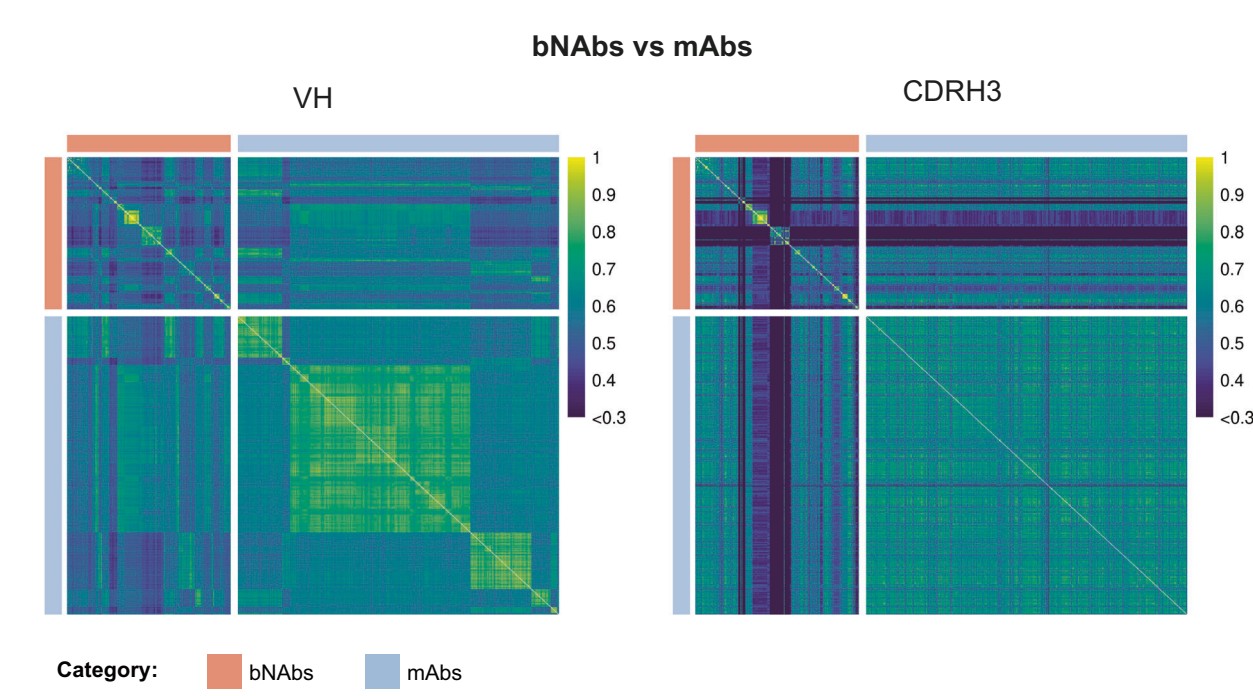

**Category:** ▮ bNAbs   ▮ mAbs

**Fig. 2 | Sequence similarity matrices of HIV-1 bNAbs and control mAbs.**
**a** Similarity matrices for 255 bNAbs grouped by antigenic site for the entire VH (left) or the CDRH3 only (right). **b** Similarity matrices of bNAbs versus mAbs with entire VH (left) and CDRH3 only (right). In the heatmaps, sequences are ranked based on their V and J genes. In both cases, matrices were created using ANARCI.

The similarity scores, ranging from 0 to 1, indicate the degree of similarity between sequences, with higher scores representing lower Levenshtein distances. **b** mAb sequences were downsampled to 500 to enable display. Source data are provided as a Source Data file.

interface or the V1V2 apex (Supplementary Fig. 3a, b). The receiver-operating characteristic (ROC) curve and corresponding AUC of 0.94 was obtained for V1V2 apex (Supplementary Fig. 3a) and 0.9 for interface (Supplementary Fig. 3b), indicating good classification performance for both antigenic sites. Furthermore, a measured AUC of 0.77 was obtained for bNAbs binding the CD4bs (Supplementary Fig. 3c).

However, the detection of bNAbs against other antigenic sites such as MPER (Supplementary Fig. 3c), and V3 loop (Supplementary Fig. 3e) was not satisfactory with an AUC close to 0.5 and 0.67, respectively.

Following this result, we allowed the DT and RF algorithms to use all available features, including VH and VL genes, and further optimized our models. We used the validation dataset to perform

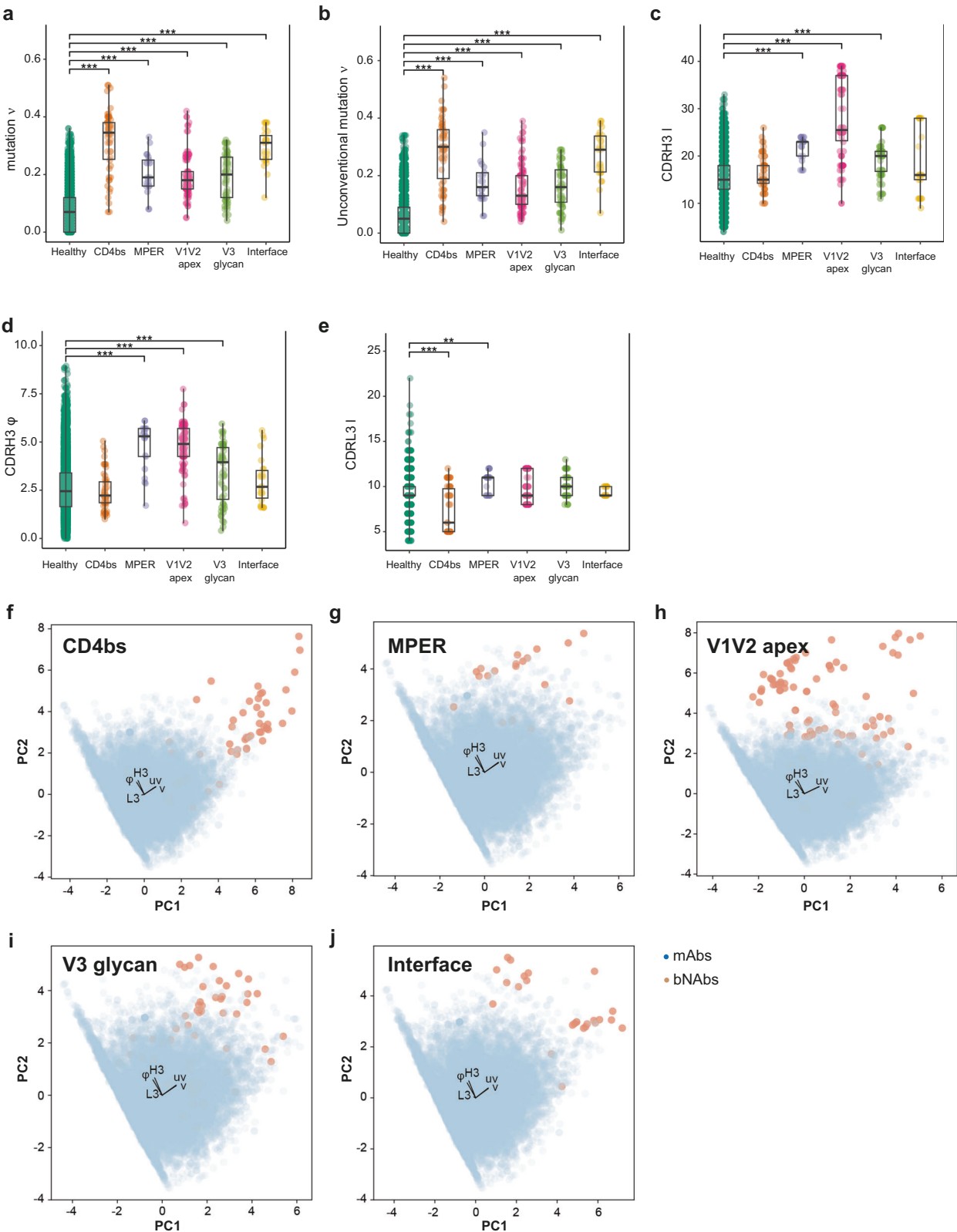

hyperparameter tuning and systematically explore different combinations of hyperparameters. We based the classifiers' hyperparameter tuning on the false positives number, and for the hierarchical model of the DT, the cost complexity pruning parameter (alpha) was set to zero (Supplementary Fig. 4). Next, entropy was chosen as the quality measurement for the split in both DT and RF (further details are presented in "Methods"). The optimal parameters for our models enabled us to

achieve an overall very good performance, with a mean AUC of 0.87 (SD = 0.11) for the DT model and 0.95 (SD = 0.08) for the RF model (Supplementary Table 2). Notably, the mean precision score was very high, reaching 1 (SD = 0) for the RF model, while it was 0.5 (SD = 0.124) for the DT model. Finally, we used the test datasets and evaluated performance metrics, including AUC, precision, recall, and accuracy for the DT and RF models (Fig. 4a, c, d). The DT algorithm exhibited

**Fig. 3 | Characteristics discriminating HIV-1 bNAbs from mAbs.** Specific properties of antibodies that allow differentiation between bNAbs and mAbs depending on the antigenic site. Shown is boxplots with center line denoting the median value (50th percentile), while the black box contains the 25th to 75th percentiles of the dataset. The black whiskers mark the 5th and 95th percentiles. **a** Mutation frequency (v), **b** unconventional mutation frequency (uv), **c** CDRH3 length (H3), **d** CDRH3 hydrophobicity (φ), and **e** CDRL3 length (L3) were statistically compared with Kruskal–Wallis's test followed by Dunn's post hoc test. Only significant comparisons with mAbs are shown, with: *$P < 0.05$, **$P < 0.01$, and ***$P < 0.005$. **f–j** Principal component analysis (PCA) of the immunoglobulins using five features (v, uv, H3, φ, and L3). The feature weight for PC1 (Principal Component 1) and PC2 (Principal Component 2) is shown by black arrows. Each bNAb category is represented by a single plot per antigenic site, **f** CD4bs, **g** MPER, **h** V1V2 apex, **i** V3 glycan, and **j** gp120/gp41 **i**nterface. For data in (**a–e**), sequences number is $n = 14,962$ for healthy, $n = 54$ for CD4bs, $n = 21$ for MPER, $n = 98$ for V1V2 apex, $n = 56$ for V3 glycan and $n = 26$ for Interface. Adjusted $P$ values with the Holm method are as follows: (**a**) healthy vs CD4bs $P = 2.04\text{e-}31$, MPER $P = 1.72\text{e-}10$, V1V2 $P = 5.94\text{e-}44$, V3 $P = 1.06\text{e-}20$ and interface $P = 3.00\text{e-}17$. **b** Healthy-CD4bs $P = 1.51\text{e-}30$, MPER $P = 4.87\text{e-}10$, V1V2 $P = 7.90\text{e-}34$, V3 $P = 3.08\text{e-}21$ and interface $P = 1.03\text{e-}16$. **c** Healthy-MPER $P = 1.04\text{e-}09$, V1V2 $P = 5.33\text{e-}45$ and V3 $P = 3.12\text{e-}10$. **d** Healthy-MPER $P = 3.97\text{e-}08$, V1V2 $P = 8.92\text{e-}36$ and V3 $P = 6.08\text{e-}04$. **e** Healthy-CD4bs $P = 8.89\text{e-}10$ and MPER $P = 0.02$. Source data are provided as a Source Data file.

superior recall and precision performance compared to the AD algorithm, while the RF algorithm demonstrated even higher performance, achieving a minimum AUC of 0.92 for all tested antigenic sites. It achieved a precision of 1 for almost all antigenic sites (0.83 for the interface). Moreover, an AUC of 1.0 and 0.95 for the MPER and interface site respectively, but also 0.95 for the V3 glycan, demonstrating that RF had the best performance as expected. Next, we reviewed the selected parameters used as RF classifiers. Interestingly, among the seven most important features, some were shared between the antigenic sites, while others were distinct. For instance, the mutation frequency and hydrophobicity of the CDRH3 were often key predictors (Fig. 4e). While the mutation frequency was an expected characteristic due to the long affinity maturation process required to obtain bNAbs, hydrophobicity of the CDRH3 might be interpreted as a consequence of the important glycan shield surrounding gp120/gp41 trimer. The frequency of unconventional mutations and length of the CDR3 light chain appears as an important feature for anti-CD4bs and is in agreement with reported bNAbs of this antigenic class[41]. The essential features associated with anti-interface bNAbs were also characterized by their mutation frequency both conventional and unconventional. The V1V2 apex binders were classified based on their CDRH3 lengths. Interestingly, bNAbs targeting the V3 glycan and MPER have a more balanced classification with features such as CDRH3 hydrophobicity, mutation, and CDRH3 length sharing similar weights (Fig. 4e). The immunoglobulin variable VH5-51 gene segment was associated with bNAbs targeting the V3 glycan as previously reported for 35% of human anti-V3 bNAbs[42]. As a final validation step, we compared the prediction results of each algorithm. These results indicate that predictors are specific to the antigenic site, even if the mutation frequency and hydrophobicity were always important.

Altogether, we observed that the different methods (AD, DT, and RF) identified the same true positives, while there was minimal overlap in false positives (Supplementary Fig. 5). To increase robustness and decrease the likelihood of false positive predictions, we combined different classifiers using the Super Learner Ensembles algorithm (SL) as an additional validation step[43]. SL is an algorithm combining multiple models to make an "ensemble" prediction. The SL algorithm exhibited very high accuracy and precision performance with a score of 1 for all antigenic sites (Supplementary Fig. 6a) and achieved high performance for the MPER, V1V2 apex, and interface antigenic sites with a minimum AUC of 0.92 (Supplementary Fig. 6b). In contrast, the AUC was lower for the CD4bs, and V3 glycan antigenic sites (0.77 and 0.68), with a recall score of 0.53 and 0.35, respectively (Supplementary Fig. 6a). Based on the performance of our machine-learning approach for the Rapid Automatic Identification of bNAbs from Immune Repertoires (RAIN), we decided to use it on experimental samples in an effort to discover new bNAbs.

## Experimental validation of the pipeline using de novo immune repertoires

To identify potential bNAbs, we investigated the neutralizing activity of purified immunoglobulin G (IgG) from the sera of different HIV-1 infected donors. Polyclonal IgGs from the serum of donors were

purified with protein G resin and tested on the global HIV-1 panel of reference strains, containing strains that are representative of the global epidemic[44,45]. Interestingly, we observed that sera of donors 3, 11 and to some extent donor 9 had a broad neutralizing activity (Fig. 5a). In contrast, sera from donors 1, 2, 5, 6, 7, and 8 were able to neutralize only one or two viral strains (Fig. 5a). Based on this result, we selected the serum of donor 3 as test sample for the bNAb identification, while sera of donors 1 and 2 were selected as negative controls. We isolated IgG-class-switched B cells from peripheral blood mononuclear cells (PBMCs) of the different donors and performed single-cell sequencing of the B-cell receptors (BCRs) (B3, G3, S4, and G4). Importantly, no enrichment step was applied for B-cell sorting to ensure an unbiased repertoire for the downstream analysis. After filtering for error-corrected and productive sequences, we successfully reconstituted a set of 15,713 IgG sequences for donor 3. As a negative control, we sequenced BCRs from IgG+ memory B cells of donors 1 and 2 (that did not have sera with broad neutralization activity), which resulted in the acquisition of 8347 IgG sequences (D1 and D2). Interrogation of the RAIN pipeline on the sequences obtained from donor 3, led to the identification of several potential bNAbs, but only 3 were recognized by the three algorithms out of 15,713 paired sequences. To assess the specificity of RAIN on HIV samples, we decided to analyze B-cell repertoires from individuals exposed to a different viral infection or post-vaccination as an alternative control. We used sequences obtained from an Influenza vaccinated donor at days 7 and 9 post-vaccination[46]. These sequences correspond to three sequencing runs of 4691, 8222, and 8052 paired BCRs sequences, respectively. While these repertoires contain Influenza bNAbs[46], our models did not detect any HIV bNAbs, indicating their specificity toward anti-HIV sequences (Supplementary Fig. 7a). To further confirm the three predicted HIV-1 bNAbs found in donor 3, we used the SL model, which identified thirteen potential bNAbs in this donor: six predicted to bind to the CD4-binding site, one to V1V2 apex, and six interface binders (Supplementary Fig. 7b). Interestingly, SL confirmed our predicted bNAbs, but also identified an anti-V1V2 apex binder in donor 2. These three potential bNAbs were constantly identified as CD4 binders (bNAb2101, bNAb4251, and bNAb1586) and belong to the VRC01 class of bNAbs (Supplementary Fig. 8).

## Binding and neutralization properties of the identified bNAbs

To consolidate these findings, we cloned the three potential bNAbs and some additional antibodies as negative control (hereafter referred to as mAbs). BNAbs and mAbs were recombinantly produced to test their specificity and neutralizing activities. We first assessed their binding to the envelope trimer SOSIP (using the clade A gp140 envelope stabilized prefusion trimer BG505 DS-SOSIP)[47,48], which is known to bind bNAbs that are representative of the majority of the known gp120 neutralizing antibody class[49,50]. Using biolayer interferometry (BLI), we detected high-affinity interactions between all the identified bNAbs and SOSIP, characterized by an apparent equilibrium dissociation constant ($K_D$) of $115 \pm 15$ nM, $3 \pm 0.6$ nM, and $0.4 \pm 0.03$ nM and for bNAbs 1586, 2101, and 4251 respectively. In contrast, no interaction could be detected between the control mAbs and SOSIP (Fig. 5b and Supplementary Fig. 9a). To further characterize these interactions, we

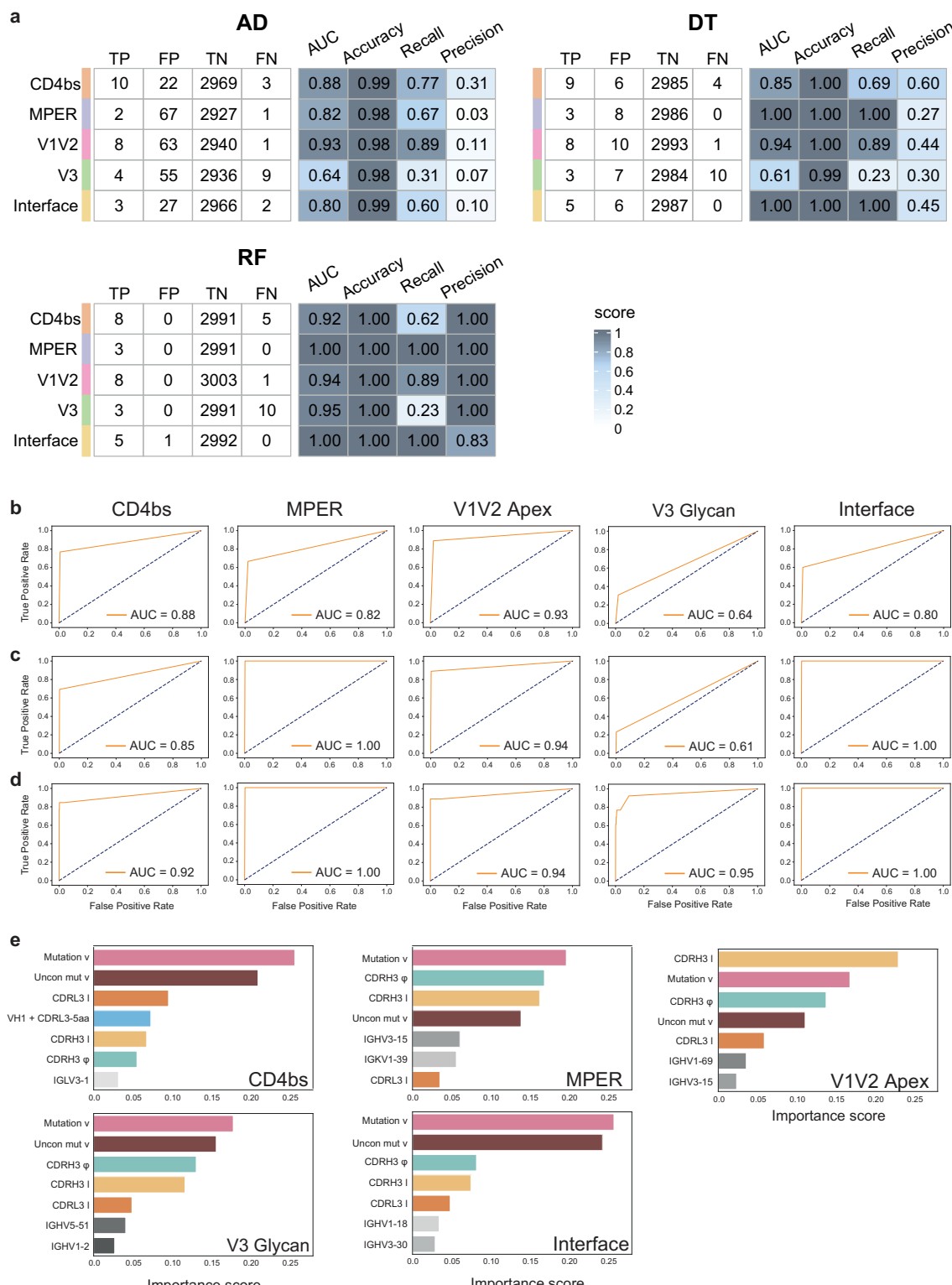

**Fig. 4 | Performance of RAIN machine-learning models. a** Performance metrics of the three algorithms using the test dataset with Accuracy = (TP + TN)/(TP + FP + TN + FN), Recall = TP/(TP + FN), and Precision = TP/(TP + FP). **b–d** Receiver-operating characteristic (ROC) curves and corresponding area under the curve (AUC) statistics for each bNAb antigenic site with the test dataset. Each row represents one algorithm, **b** AD, **c** DT, and **d** RF. **e** Most important features with their scores for each bNAb classified by binding antigenic site using the Random Forest classifier.

calculated the affinity of the fragment antigen binding (Fab) to SOSIP trimers and obtained a $K_D$ 5 ± 2.4 nM, and 17.5 ± 4 nM for Fab4251 and Fab2101, respectively (Supplementary Fig. 9b). Of note, Fab1586 demonstrated poor affinity with a $K_D$ measure of 1 μM (Supplementary Fig. 9b). To investigate the neutralization potency of these bNAbs, we

sought to determine their $IC_{50}$ using the global HIV-1 panel strains on TZM-bl cells[44,45]. We observed a broad neutralization activity across tiers and viral clade for bNAb4251, with a geometric mean $IC_{50}$ of 1.8 μg/ml (Fig. 5c). Moreover, bNAb2101 could also neutralize different HIV strains and more specifically clade AE viruses, however, its

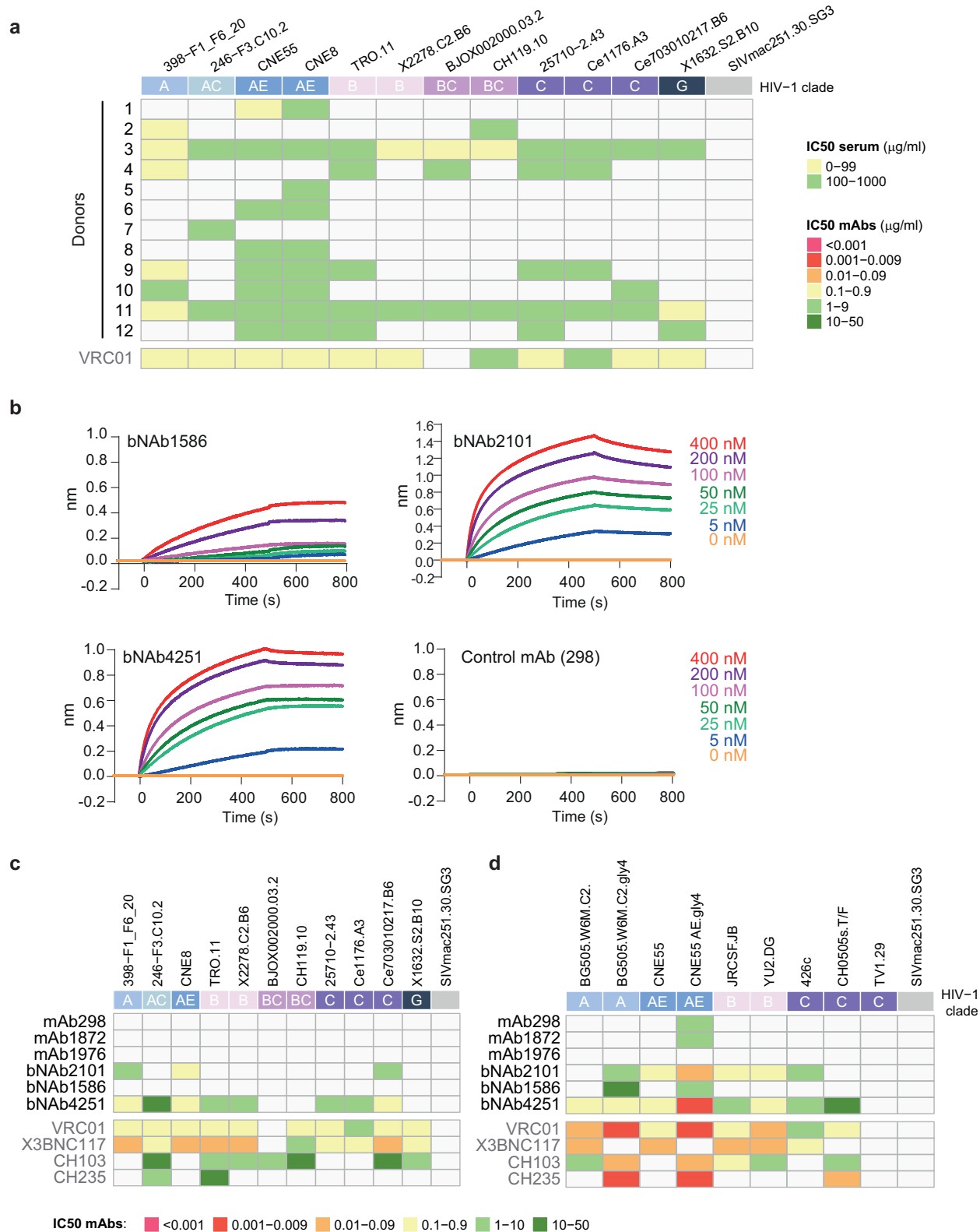

neutralization profile could not be considered broad (Fig. 5c). In contrast, bNAb1586 was a relatively poor neutralizer, only able to inhibit the CNE55 strain at 38 μg/ml (Fig. 5d). Importantly, none of the antibodies had an effect on the SIVmac251.30.SG3 virus indicating a specific neutralization activity. Overall, bNAb4251 could neutralize about 80% of the tested viruses but was not active against the TV1.29 and BJOX002000, similar to VRC01, which targets the CD4-binding site[51].

Since both potential bNAbs were predicted to target the CD4-binding site, we further tested their neutralization potential on virus strains lacking the glycosylation surrounding the CD4bs such as the BG505.W6M.C2 strain with residues T332N (C2) or N197, N276, N363, and N462 (gly4) and other mutations previously described[52] (Fig. 5d). Finally, clade C strains were also used, since the glycan at position 362 is naturally absent. The neutralization profile showed an increased

**Fig. 5 | HIV Env binding and neutralization assays of serum and IgG samples. a** Neutralization assays were performed against 12 viruses from clades A, AC, AE, B, BC, C, and G of tiers 2. The colors of the heatmap correspond to the IC$_{50}$ of the sera in micrograms per ml. The SIVmac251.30.SG3 virus is used as a negative control. **b** Antibody–SOSIP interactions were determined by biolayer interferometry (BLI). The mAbs or bNAbs were loaded on a protein A biosensor, dipped into a solution of the SOSIP trimer at different concentrations (ranging from 5 to 400 nM), and the nm shift was recorded. BLI sensorgrams are representative examples of experiments repeated two times ($n > 2$). **c, d** Neutralization assays were performed against twelve viruses from clades A, AC, AE, B, BC, C, and G of tiers 2. **c** The colors of the heatmap correspond to the IC$_{50}$ in micrograms per ml, for each antibody. The SIVmac251.30.SG3 virus is used as a negative control. **d** Neutralization assays were performed against glycan-mutated viruses to support epitope mapping to the CD4-binding site. Neutralization assay experiments were repeated two times ($n > 2$). Source data are provided as a Source Data file.

potency specifically for the glycan mutations surrounding CD4bs, suggesting again that these antibodies target the CD4bs (Fig. 5d).

### Binding mode of Fab4251 and Fab2101 to BG505 DS-SOSIP complex

Based on the affinity and neutralization potency of Fab2101 and Fab4251, we decided to investigate their binding mode using electron microscopy. We incubated BG505 DS-SOSIP with either 3 molar excesses of Fab2101 or Fab4251 and imaged the complex after 30 min of incubation at room temperature. We used single-particle negative stain electron microscopy (nsEM) to assess the sample purity and to map antibody epitopes on the viral glycoproteins[53] (Fig. 6a, b). Particles were picked from raw micrographs, stacks were created, followed by a reference-free 2D classification. SOSIP complexes appear as homogeneous trimers, as described previously[54]. We identified that both Fabs bound to the soluble trimers in a manner similar to CD4bs-directed bNAbs, approaching the gp120 protomers from the side. To understand the molecular mechanism of the broad neutralization capacity by bNAb4251, we decided to perform cryo-EM of the Fab4251 in complex with the soluble native-like trimer BG505 DS-SOSIP[55]. After several rounds of 2D and 3D classification (Supplementary Fig. 10), we could segregate SOSIP trimers with zero, or one Fab bound. We solved the structure of the complex at a resolution of 3.8 Å (Fig. 6c and Supplementary Table 3). As predicted by RAIN, Fab4251 interacts with the CD4bs of the trimer and makes multiple contacts with both heavy and light chains (Fig. 6c, d). In total, fifty-one residues of the Fab interact with fifty-six residues on gp120, to bury a surface area (bsa) of 950 Å². The interaction is principally dictated by the heavy chain with 700 Å² bsa, while the light chain buries 250 Å² of the gp120 surface (Fig. 6d). The CDRH2 makes most of the contact, totaling a bsa of 528 Å², a binding mode similar to the previously described interaction of the CD4 receptor with gp120 (Fig. 6g). The previously solved interaction of CD4 with gp120 revealed that two amino acids, F43 and N59 of CD4, make multiple contacts centered on residues N368, E370, and W427 of gp120[56–58] (Fig. 6g). Interestingly, H54 of CDRH2 seems to mediate similar interactions with amino acids of the "F43 cavity" located at the interface between the inner and outer gp120 domains (Fig. 6g). Previously reported bNAbs targeting the CD4bs have been classified into two groups based on their mode of recognition, the VRC01 class (3BNC117, N6, N49P7, 3BNC60, VRC-PG20, NIH45-46, VRC-CH31, and 12A12) and the non-VRC01 classes (CH103, 8ANC131, VRC13, and VRC16)[59]. Structural investigations revealed that Fab4251 possesses an angle of approach similar to VRC01 (Fig. 6h), a result in agreement with its CDRH2-mediated contact on gp120, indicating that it belongs to the same antibody class (Fig. 6h). The light chain also participates in the interaction with the 5-residue LCDR3 QxxEx motif and a deletion in CDRL1 to accommodate the gp120 N276-glycan[28], a feature associated with VRC01-class antibodies.

### Discussion

The advent of single-cell technologies resulted in the growing availability of paired full-length variable heavy and light-chain BCR sequences. Therefore, immune repertoire sequencing coupled to artificial intelligence holds great promise to improve diagnosis and treatment for numerous immune-related or infectious diseases[60]. The identification of specific sequences involved in an immune response has already been successfully used in research settings to elucidate the role of immune dysregulation in conditions such as systemic lupus erythematosus, rheumatoid arthritis, type 1 diabetes, multiple sclerosis, Grave's disease, Crohn's disease, and many others[61]. However, limitations exist and only a few studies examined the benefit of incorporating full-length variable regions from heavy and light-chain sequences to predict antibody specificity. Those studies are based on sequence-based embedding models[62,63]. Other efforts have focused on finding amino acid sequence similarity to an already known antibody. The similarity approaches led to important scientific and medical discoveries[64–66], but hold some limitations when the sequences are very divergent.

In this study, we present RAIN, a pipeline based on two innovative technologies, single-cell BCR sequencing and machine learning to identify bNAbs against HIV-1, based on their binding site. Our approach differs from other methods as the parameters required for the identification are derived from selected characteristics, that are inferred from the amino acid sequences using Immcantation annotations. We demonstrate that five specific characteristics were sufficient to separate bNAbs from mAbs (non-bNAbs) into two distinct clusters within each category of antigenic sites. In addition, we identify the frequency of unconventional mutations as a key factor to define HIV-1 bNAbs. Former studies reported the presence of mutations in the frameworks of bNAbs and correlated them with the binding affinity to the CD4bs[35,67]. Our results suggest that these mutations are important characteristics for all bNAbs. This can be interpreted as a consequence of the time needed for the maturation process of bNAbs or as a modification of the immune system in response to chronic infection.

Performing a PCA analysis across all five antigenic sites, we observed that despite their sequence divergences, the weights of the features exhibited striking similarities. This result could be interpreted as an additional level of immune escape that was not studied yet[68,69]. The RAIN approach can achieve a precision of 1 for almost all antigenic sites and can be applied easily on any immune repertoire or already isolated antibody sequences to identify HIV-1 bNAbs. To our knowledge, this study pionner in silico identification of specific antibodies, that could not have been identified by sequence alignment. Importantly, another distinct aspect of our work is the experimental validation with de novo data. Data were corroborated by functional cloning, expression and purification of the antibodies, and functional neutralization assays. Moreover, we characterized the bNAb4251 binding to DS-SOSIP at almost atomic resolution using cryo-EM. In summary, our approach offers an innovative, straightforward method to search and identify antibodies in immune repertoires, accelerate antibody discovery, and might shed light on potentially unexplored mechanisms of HIV-1 immune escape.

## Methods
### Ethics statement
The research complies with all relevant ethical regulations and informed consent was obtained by all study participants ($n = 25$, 16 females and 9 males). Study protocols were approved by the Ethik-komission beider Basel (EKBB; Basel, Switzerland; reference number 342/10), the Ifakara Health Institute Institutional Review Board (Reference number IHI/IRB/No. 24-2010), and the National Institute for Medical Research (NIMR; Dar es Salaam, United Republic of Tanzania; reference number NIMR/HQ/R.8a/Vol.IX/1162).

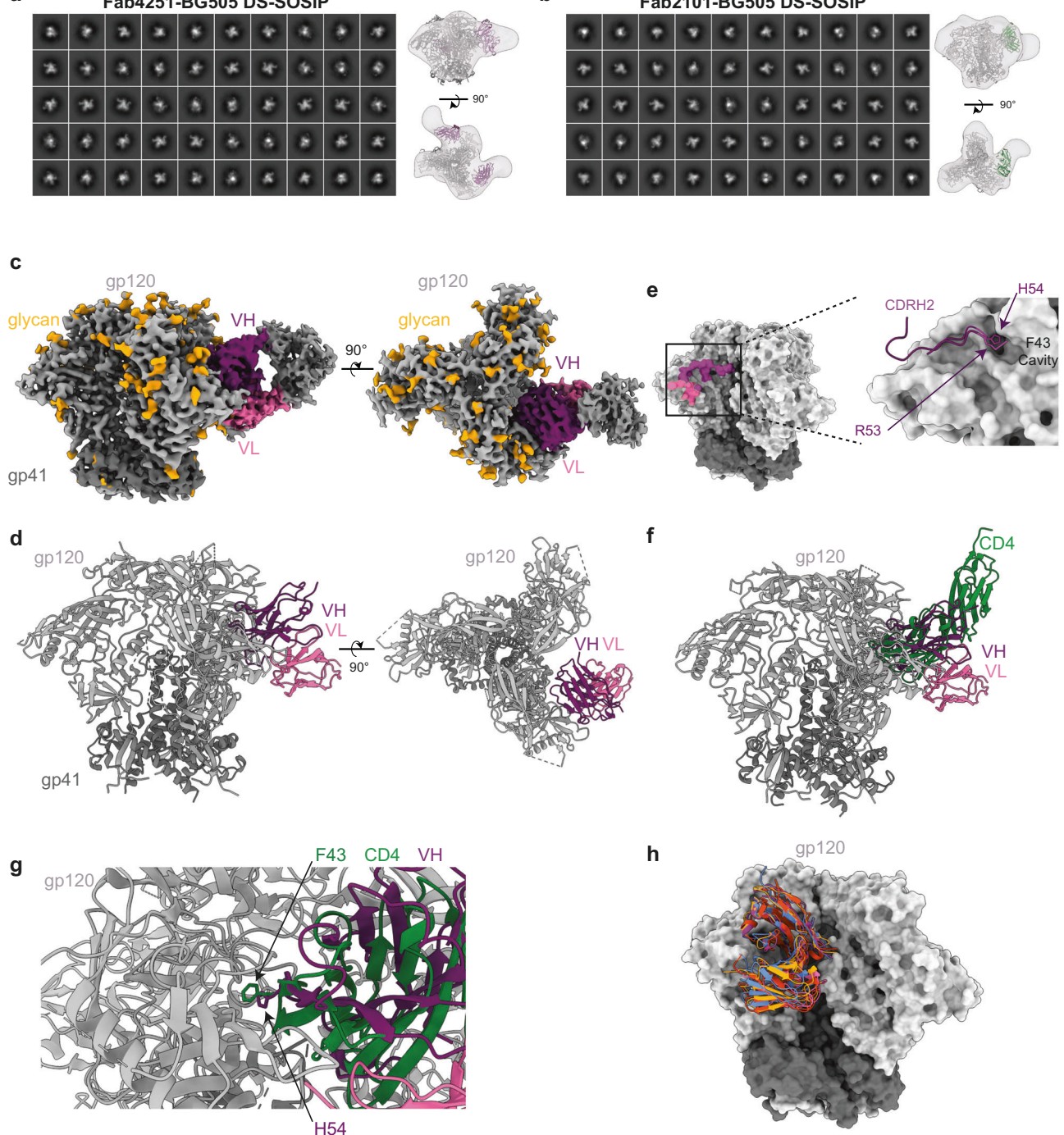

**Fig. 6 | Fab4251 and Fab2101 interaction with BG505 DS-SOSIP. a** 3D reconstruction of Fab4251-SOSIP complex by nsEM. **b** 3D reconstruction of Fab2101-SOSIP complex by nsEM. **c** Side and top views of the cryo-EM density map of the Fab4251-DS-SOSIP complex, with gp120 in light gray, gp41 in dark gray, VH in violet, and VL in pink. **d** Atomic model of Fab4251-DS-SOSIP complex shown in cartoon representation. **e** Footprint representation of the heavy and light-chain binding surface on DS-SOSIP, colored according to (**c**). Inlet on the right represents the CDRH2 loop in violet, with H54 in the F43 cavity. **f** Overlay of CD4 receptor (green) bound to SOSIP (PDB.5U1F) and Fab4251 (violet). **g** Close view of VH H54 from Fab4251 and F43 in CD4. **h** Overlay of VRC01-class antibodies on SOSIP with Fab4251 (violet), VRC01 (PDB.6V8X, green), PG04 (PDB.4I3S, red), and 3BNC60 (PDB.4GW4, orange).

## Serum IgG isolation

Serum samples from HIV-1-infected individuals were incubated with Protein G Sepharose (GE Life Sciences) 4 °C for 1 h. IgGs were eluted from chromatography columns using 0.1 M glycine (pH = 2.9) into 0.1 M Tris (pH = 8.0)[70]. Samples were run through Zeba Spin Desalting Columns 7 K MWCO (Thermo Scientific, 89882) Concentrations of purified IgGs were determined by UV/Vis spectroscopy (A280) on a Nanodrop 2000 and samples were stored at −80 °C.

## B-cell sorting

The CD19+ cell fraction was enriched from PBMCs by positive selection with CD19 magnetic microbeads (Miltenyi Biotech) and subsequently

stained on ice for 30 min with the following fluorochrome-labeled mouse monoclonal antibodies: CD20-PE-Cy7 (dilution 1:50, clone L27, catalog no. 335793, BD Biosciences) and F(ab')2-Goat anti-Human IgG Fc secondary antibody, APC (dilution 1:100, RRID: AB_2337695, Jackson ImmunoResearch). Cells were sorted to over 98% purity on a FACS Aria III (BD) using the following gating strategy: circulating memory B cells were sorted as CD20+ IgG+ cells. FACS-sorted cells were collected in 6 μl FCS in Eppendorf tubes that were pre-coated overnight with 2% BSA.

## Single-cell BCR-seq library preparation and sequencing

**10X Genomics.** The 5′ single-cell VDJ libraries were generated using Chromium Next GEM Single Cell V(D)J Reagent kit v.1, 1.1 or v.2 (10X Genomics) according to the manufacturer's protocol. Paired heavy and light-chain BCR libraries were prepared from the sorted B-cell populations. Briefly, up to 20,000 memory B cells per well of 10X chip were loaded in the 10X Genomics Chromium Controller to generate single-cell gel beads in emulsion. After reverse transcription, gel beads in emulsion were disrupted. Barcoded complementary DNA was isolated and used for the preparation of BCR libraries. All the steps were followed as per the manufacturer's instructions in the user guide recommended for 10X Genomics kit v.1, 1.1, or 2. The purified libraries from each time point were pooled separately and sequenced on the NextSeq550 (Illumina) as per the instructions provided in 10X Genomics user guide for the read length and depth.

**BD rhapsody.** Memory B cells were targeted for single-cell targeted RNA-seq and BCR-Seq analysis using the BD Rhapsody Single-Cell Analysis System[71] (BD Biosciences). Briefly, the single-cell suspension was loaded into a BD Rhapsody cartridge with >200,000 microwells, and single-cell capture was achieved by random distribution and gravity precipitation. Next, the bead library was loaded into the microwell cartridge to saturation so that the bead was paired with a cell in a microwell. The cells were lysed in a microwell cartridge to hybridize mRNA molecules onto barcoded capture oligos on the beads. These beads were then retrieved from the microwell cartridge into a single tube for subsequent cDNA synthesis, exonuclease I digestion, and multiplex-PCR–based library construction. Sequencing was performed on NovaSeq paired-end mode.

**Singleron.** Single-cell suspensions with $1 \times 10^5$ cells/mL in PBS were prepared. Then, the suspensions were loaded onto microfluidic devices, and scRNA-seq libraries were constructed according to the Singleron GEXSCOPE protocol in the GEXSCOPE Single-Cell RNA Library Kit (Singleron Biotechnologies)[72]. Individual libraries were diluted to 4 nM and pooled for sequencing. Pools were sequenced on an Illumina HiSeq X with 150 bp paired-end reads.

## Recombinant antibody production

Expi293 cells (Thermo Fisher Cat No. A14527) were diluted to a final volume of 0.5 L at a concentration of $2.5 \times 10^6$ cells mL$^{-1}$ in Expi293 media[73]. Heavy-chain and light-chain plasmids were complexed with Polyethyleneimine (Thermo Fisher) and added to the cells. On day five, cells were cleared from cell culture media by centrifugation at $10,000 \times g$ for 30 min, and the supernatant was subsequently passed through a 0.45-μm filter. The supernatant containing the recombinant antibody was purified with the HiTrap Protein A HP column (Cytiva, 17040301) on the Äkta pure system (Cytiva). The resin was washed with 75 mL of phosphate-buffered saline (PBS). A total of 25 mL of 0.1 M glycine pH 2.9 were used to elute the antibody from the protein A resin. The acidic pH of the eluted antibody solution was increased to ~7 by the addition of 1 M Tris pH 8.0. The antibody solution was buffer exchanged to PBS by the HiPrep 26/10 Desalting column (GE Healthcare) or Size Exclusion Chromatography Superdex 16/600 HiLoad (Cytiva), filtered, snap-frozen in liquid nitrogen, and stored at −80 °C.

## Fragment antigen binding (Fab) generation

For the Fab production, the heavy chain was engineered with a two amino acids glycine serine linker followed by a six-histidine tag and stop codon. Light and mutated heavy chains were transfected as described in the previous section. Cell supernatant was harvested five days post- transfection and purified by IMAC chromatography (HisTrap excel, Cytiva) using the elution buffer 25 mM Tris pH 7.4, 150 mM NaCl, 500 mM imidazole. The eluat was buffer exchanged to 25 mM Tris pH 7.4, 150 mM NaCl, 0.085 mM n-dodecyl β-D-maltoside (DDM) on a HiPrep 26/10 Desalting column (GE Healthcare), followed by Size Exclusion Chromatography on a Superdex 16/600 HiLoad column (Cytiva)[74]. The sample was concentrated using an Amicon filter 10 kDa cutoff, snap-frozen, and stored at −80 °C until further use.

## Recombinant HIV-1 envelope SOSIP gp140 production

BG505 DS-SOSIP trimer[75] production and purification were performed as previously described[48]. Briefly, prefusion-stabilized Env trimer derived from the clade A BG505 strain was stably transfected in CHO-DG44 cells and expressed in ActiCHO P medium with ActiCHO Feed A and B as feed (Cytiva). Cell supernatant was collected by filtration through a Clarisolve 20MS depth filter followed by a Millistak + F0HC filter (Millipore Sigma) at 60 LMH. Tangential Flow Filtration was used to concentrate and buffer exchange clarified supernatant in 20 mM MES, 25 mM NaCl, pH 6.5. The trimer was then purified by ion exchange chromatography as described[48]. Fractions containing the BG505 DS-SOSIP protein were pooled, sterile-filtered, snap-frozen, and stored at −80 °C.

## IgG neutralization assay

Neutralization assays with IgGs against the 12-strain "global" virus panel, were performed in 96-well plates as previously described[44,76,77]. Briefly, 293T-derived HIV-1 Env-pseudotyped virus stocks were generated by cotransfection of an Env expression plasmid and a pSG3ΔEnv backbone. Animal sera were heat-inactivated at 56 °C for 1 h and assessed at 8-point fourfold dilutions starting at 1:20 dilutions. Monoclonal antibodies were tested at 8-point fivefold dilutions starting at 50 μg/ml or 500 μg/ml. Virus stocks and antibodies (or sera) were mixed in a total volume of 50 μL and incubated at 37 °C for 1 h. TZM-bl cells (20 μl, 0.5 million/ml) were then added to the mixture and incubated at 37 °C. Cells were fed with 130 μL cDMEM on day 2, lysed, and assessed for luciferase activity (RLU) on day 3. A nonlinear regression curve was fitted using the 5-parameter hill slope equation. The 50% and 80% inhibitory dilutions ($ID_{50}$ and $ID_{80}$) were determined for sera and the 50% and 80% inhibitory concentrations ($IC_{50}$ and $IC_{80}$) were determined for mAbs. All samples were tested in duplicates.

## Biolayer interferometry

The biolayer interferometry experiments using SOSIP were performed as follows. All experiments were performed in reaction buffer (TBS pH 7.4 + 0.01% (w/v) BSA + 0.002% (v/v) Tween 20) at 30 °C using an Octet K2 instrument (ForteBio). Protein A (Fortebio) biosensor probes were first equilibrated in reaction buffer for 600 s. IgGs were diluted to 5 μg/ml in reaction buffer and immobilized onto the protein A probes for 300 s, followed by a wash for 300 s in reaction buffer. The binding of SOSIP trimers to the IgGs was then measured at various concentrations for 500 s, followed by dissociation for 300 s in reaction buffer. Analysis was performed using the Octet software with bivalent analyte fitting for antibody binding and 1.1 analyte fitting for the interaction with Fabs. Association and dissociation curves are visualized by GraphPad Prism version 9.0.

## Negative stain electron microscopy

The samples were adsorbed to a glow-discharged carbon-coated copper grid 400 mesh (EMS, Hatfield, PA, USA), washed with deionized water, and stained with a 1% uranyl acetate solution for 20 s.

Observations were made using an F20 electron microscope (Thermo Fisher, Hillsboro, USA) operated at 200 kV[73]. Digital images were collected using a direct detector camera Falcon III (Thermo Fisher, Hillsboro, USA) 4098 × 4098 pixels. Automatic data collection was performed using the EPU software (Thermo Fisher, Hillsboro, USA) at a nominal magnification of ×62,000, corresponding to a pixel size of 1.65 Å using a defocus range from −1 μm to −2.5 μm. Image pre-processing, two-dimensional classification, and three-dimensional processing was done using the CryoSPARC software (Version 4.4)[78].

## Cryo-EM sample preparation
BG505 DS-SOSIP trimers complexes were prepared using a stock solution of 5 mg/ml trimer incubated with a threefold molar excess of bNAb4251 for 10 min. To prevent aggregation and interaction of the trimer complexes with the air-water interface during vitrification, the samples were incubated in 25 mM Tris pH 7.4, 150 mM NaCl, 0.085 mM DDM. Samples were applied to plasma-cleaned QUANTIFOIL holey carbon grids (EMS, R2/2 Cu 300 mesh). The grid was plunge frozen using a Vitrobot MarkIV (Thermo Fisher, Hillsboro, USA)with humidity and temperature control.

## Cryo-EM data collection
Grids were screened for particle presence and ice quality on a TFS Glacios microscope (200 kV), and the best grids were transferred to a TFS Titan Krios G4. Cryo-EM data were collected using a TFS Titan Krios G4 transmission electron microscope, equipped with a Cold-FEG on a Falcon IV detector in electron counting mode. Falcon IV gain references were collected just before data collection. Data were collected using TFS EPU v2.12.1 utilizing the aberration-free image shift protocol, recording four micrographs per ice hole. Movies were recorded at a magnification of ×165,000, corresponding to the 0.73 Å pixel size at the specimen level, with defocus values ranging from −0.9 to −2.4 μm. Exposures were obtained with 39.89 e⁻ Å⁻² total dose, resulting in an exposure time of ~2.75 s per movie. In total, 15,163 micrographs in EER format were collected.

## Cryo-EM data processing and structure fitting
Data processing was performed with cryoSPARC (Version 4.4) including Motion correction and CTF determination[78]. Particle picking and extraction (extraction box size 350 pixels²) were carried out using cryoSPARC Version 4.4[78]. Next, several rounds of reference-free 2D classification were performed to remove artifacts and selected particles were used for ab initio reconstruction and hetero-refinement. After hetero-refinement, 72'497 particles contributed to an initial 3D reconstruction of 3.8 Å resolution (Fourier-shell coefficient (FSC) 0.143) with C1 symmetry. A model of a SOSIP trimer (PDB ID 4TVP)[79] or AlphaFold2 (ColabFold implementation) models of the 4251 Fab were fitted into the cryo-EM maps with UCSF ChimeraX (Version 1.5). These docked models were extended and rebuilt manually with refinement using Coot (Version 0.9.8.8) and Phenix (Version 1.21)[80,81]. Figures were prepared in UCSF ChimeraX, and Pymol (Version 4.6)[82]. The numbering of Fab4251 is based on the Kabat numbering of immunoglobulin models[83]. Buried surface area measurements were calculated within ChimeraX and PISA[84].

## CATNAP sequences
For all antigenic sites, paired bNAb sequences were collected from the CATNAP database[32] as of January 1, 2022 as nucleotide and amino acid sequences. First, the 249 heavy-chain and 240 light-chain nucleotide sequences were annotated with Igblastn[36]. Sequences were then processed and analyzed using the Immcantation Framework (http://immcantation.org) with MakeDB.py from Change-O v1.2.0 (with the options --extended –partial). Next, bNAbs were filtered by a dedicated Java script to keep only sequences with an annotated CDR3 and paired

sequences (VH + VK/L). Each paired antibody was associated with its targeting Env antigenic site, information provided by the database CATNAP text file (abs.txt as of January 1, 2022). The 27 CATNAP antibodies with only the protein sequences available were annotated with IgBlastp followed by MakeDB.py from Change-O v1.2.0 (with the options igblast-aa --extended). In parallel, using the fasta protein sequences, ANARCI[85] was used to identify the junction region. As for nucleotide sequences, paired and annotated CDR3 bNAbs were filtered in. In total, 255 bNAbs sequences were collected. Repartition of the antigenic site is as follows: 54 bNAbs target the CD4bs, 21 MPER, 98 V1V2, 56 V3, and 26 interface.

## Paired B-cell receptor repertoires
For the training and evaluation of the machine-learning models, paired BCR repertoires of ten healthy donors were collected. The repertoires were obtained from various sources (Supplementary Data Files 1) and sequenced using 10X genomics technology. Annotation and processing of the sequences were done as previously described[39] and resulted in the generation of a customized AIRR format table containing 14,962 paired BCRs. For HIV-1 immune donors three different sequencing technologies were employed: 10X genomics (D1, D2, G3, and G4), Singleron (S4), and BD Rhapsody (B3). Single-cell sequencing of selected HIV-1 immune donors using Singleron technology was processed using celescope v1.14.1 (https://github.com/singleron-RD/CeleScope) with "flv_CR" mode utilizing cellranger v7.0.1. BD rhapsody single-cell sequencing was first processed using BD Rhapsody Targeted mRNA Analysis Pipeline (version 1.11) and then, using a custom script, the generated "VDJ_Dominant_Contigs.csv" file was converted into cellranger-like output files, namely filtered_contig_annotations.csv and filtered_contig.fasta. Lastly, the 10X Genomics single-cell sequencing was processed with cellranger v7.0.1. The cellranger output files of the different HIV-1 repertoires enabled us to annotate and process them as described earlier, resulting in a table of paired BCRs with AIRR characteristics. The six different experiments resulted in 2152 BCRs for D1, 6195 BCRs for D2, 4008 BCRs for B3, 3794 BCRs for G3, 3112 BCRs for S4, and 4799 BCRs for G4.

## Sequence similarity matrices
All mAbs and bNAbs VDJ protein sequences were initially aligned using ANARCI with IMGT format. Subsequently, employing a custom R script, two similarity matrices were generated: one encompassing the entire VDJ sequence (VH) and the other focusing solely on the CDRH3 region. For each pair of sequences, a Levenshtein distance was computed, yielding a similarity score ranging from 0 to 1 (higher score representing lower Levenshtein distance). Heatmaps were constructed with the pheatmap R package, to visualize the following comparisons: all five antigenic site categories of bNAbs and the comparison bNAbs versus mAbs (mAbs sequences were downsampled to 500 sequences). Sequences were ranked based on their V and J genes.

## Data preprocessing
Using a custom script, AIRR characteristics were converted into our features of interest. The "mutation frequency" was calculated using the difference of residues between the protein sequence of the BCR and its germline sequence in the FWR1 + CDR1 + FWR2 + CDR2 + FWR3 regions (VH gene). The "framework mutation frequency" was calculated similarly but using only FWR1 + FWR2 + FWR3. The "hydrophobicity" of the CDRH3 sequences was computed using a customized score, with aromatic residues having the highest value (1 for W, 0.75 for Y, and 0.5 for F). Residues A, L, I, M, P, and V were set to 0.1, while the rest of the resides were set to zero. The values of all residues were summed up for each CDRH3. In addition, the length of the CDRH3, CDRL3, VH, and VL/K genes were considered

as features. Two extra features were added to be used by the anomaly detection algorithm: "VH1 + CDRL3 length of five residues" with a zero or one value designed for the bNAbs targeting the CD4bs and "VH1-69 + VK3-20 + GW motif in the CDRH3" with a zero or one value for the bNAbs targeting MPER.

### Training and evaluation of machine-learning models

Three ML-based approaches were trained on the features table generated using BCRs obtained from healthy donors and bNAbs datasets, using Python v3.8.16 and scikit-learn v1.0.2. These algorithms were: Anomaly Detection (AD), Decision Tree (DT), and Random Forest (RF). For each antigenic site, the dataset was partitioned into training, validation, and test sets with a 60:20:20 ratio, setting random.seed to 1 for all models. For the AD model, bNAbs data were removed from the training set, since this algorithm only trains with non-anomaly data. For this model, the features with discrete values were first normalized using the preprocessing.normalize method (axis=0) from the scikit-learn library. Features exhibiting significantly different values from the normal distribution were selected for each antigenic site, which included the frequency of mutations in the V genes and in the frameworks. For CD4bs, we added the combined feature VH1 + CDR3L with a length of five residues. For MPER, we included the combined feature VH1-69, VK3-20, and the GW motif in CDRH3. In addition, CDRH3 hydrophobicity was added for MPER, V1V2, and V3. Lastly, CDRH3 length was incorporated for V1V2 and V3. Using the validation test, a multivariate normal random variable was calculated with the mutivariate_normal function from the scipy package v1.8.0 and used for setting the optimal Epsilon parameter ($\varepsilon$) minimizing the false positive numbers. The Epsilon value was set to 619.55 for CD4bs, 231501.41 for MPER, 866803.64 for V1V2, 845445.99 for V3, and 24.36 for interface. Those threshold values were used on the test set to predict a BCR as an anomaly (bNAb) or not. For DT and RF models, V genes (for heavy and light chains) were one-hot encoded as a preprocessing step, resulting in a total of 122 features in the features table. Hyperparameter tuning was conducted using the validation dataset, minimizing the number of false positives. DT models were trained with a balanced class weight, the Entropy criterion for measuring the quality of splits, and the cost complexity pruning parameter alpha of zero. RF models were trained with 100 estimators, a balanced class weight, the Entropy criterion for measuring the quality of splits, maximum samples were set to 1.0, maximum depth of tree of "none", maximum features of 11 ($\sqrt{122}$), and bootstrapping to build trees. Matplot library v3.6.2 was used to generate ROC plots from performance results and to generate the Venn diagrams showing the intersection of the number of true positives or false positives between the three models. The Super Learner Ensembles algorithm was implemented using the ML-Ensemble (mlens) v0.2.3 library. For each antigenic site, the dataset was partitioned into train and test sets with a 75:25 ratio. The Super Learner was created with the precision score as scorer parameter, a k-fold cross-validation of ten folds, and the option shuffle set to true. The following classifiers were used as based models in the Super Learner algorithm: DecisionTreeClassifier, SVC (Support Vector Classification), KNeighborsClassifier, AdaBoostClassifier, BaggingClassifier, RandomForestClassifier, and ExtraTreesClassifier. A LogisticRegression was used as the meta-model, with the solver parameter set to "lbfgs".

### Statistical analysis

Flow cytometric data were acquired using BD FACSDiva (v.9.0) software. Flow cytometric data were analyzed using FlowJo (v.10.7.1). Statistics were conducted using R Statistical Software (v4.2.1) and ggstatsplot package[86]. The Complex Heatmap package was used for visualization[87]. No statistical methods were used to predetermine the sample size. The experiments were not randomized, and investigators were not blinded to allocation during experiments and outcome assessment.

### Reporting summary

Further information on research design is available in the Nature Portfolio Reporting Summary linked to this article.

## Data availability

The raw sequencing data files for single-cell VDJ sequencing generated in this study have been deposited in the GEO database: GSE229123. Cryo-EM map generated in this study have been deposited on EMDB: EMD-19665, with PDB accession number 8S2E. All other data supporting the findings of this study are available from the corresponding author on request. Source data are provided with this paper.

## Code availability

The complete workflow and associated scripts are available on https://github.com/MathildeFogPerez/manuscript-bnab-foglierini. A set of instructions on how to use the workflow and completely reproduce the results shown herein is available there.

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

## Acknowledgements

The authors thank the study participants in Tanzania for donating blood samples for these studies. Sample collection was funded through this work is part of the IDEA project "Dissecting the Immunological Interplay between Poverty Related Diseases and Helminth infections: An African-European Research Initiative" (https://ec.europa.eu/research/health/infectious-diseases/neglected-diseases/projects/014_en.html) supported by the European Commission under the Health Cooperation Work Program of the 7th Framework Program (Grant 241642). The authors acknowledge the Vaccine Research Center, National Institute of Allergy and Infectious Diseases, National Institutes of Health, Bethesda, Maryland, USA for BG505 DS-SOSIP trimer; Alison Lin, David Vinyals Sales and Craig Fenwick from Lausanne University Hospital for advice and preliminary experiments with SOSIP trimer; Lausanne Genomic Technologies Facility, UNIL for next-generation sequencing of some of the samples and David Kalbermatter from the Dubochet Center for Imaging, Bern for cryo-EM grids data collection. This study was supported by the Swiss National Foundation (Grant number: 310030_20467) and intramural funding from Lausanne University—Lausanne University Hospital to L.P. Figure 1, created with BioRender.com, released under a Creative Commons Attribution-NonCommercial-NoDerivs 4.0 International license.

## Author contributions

M.F. and L.P. designed the project. P.N. with help from R.S. and R.R.W. performed and analyzed the experiments M.F., P.J., and R.G. computational work. S.O.D. and N.A.D.R. performed and analyzed the pseudo-viral neutralization assay experiments. D.D. set up of cryo-EM condition. M.M., O.L., C.D., Y.D.M., C.P., and M.P. samples or reagents. L.P. conceptualization, supervision, study design, data interpretation, and resources.

## Competing interests

The authors declare no competing interests.
