## [Peer Review File · Nature Communications]

Reviewers' Comments:

Reviewer #1:

Remarks to the Author:

In this article, the authors developed a machine-learning approach for HIV-1 bnab prediction.

Briefly, the authors used a dataset of HIV-1 bnabs as well as healthy control data to delineate a classifier for HIV-1 bnab prediction.

While this study is of interest and the question is certainly interesting, I have a several comments regarding the approach and the claims made.

Major comments

- The authors claim that their ML approach predicts HIV-1 bnab. I am not so sure about that. As shown by the authors, the bnab seqs are very different from normal antibodies in terms of SHM amount, for example. For me to believe the authors, they would have to show that their algorithm is not only specific versus healthy controls but also versus other infections. For example, if the authors applied their approach to influenza data, their approach should not find any HIV-1 bnabs, although there might be influenza bnabs in the data. In other words, I don't think their approach has learned what a HIV-1 bnab is. Their approach has probably only learned what a very strange looking antibody sequence vs normal control antibodies looks like. Can the authors comment on that?

- I don't understand why the authors need to use 4 different classifiers? what do they each add? and what features are the most important for classifying bnab vs non-bnab?

- how similar are train and test data sequence-wise?

- How similar are bnab and control data sequence-wise?

- The catnap dataset is quite small, yet the accuracy is quite high (which is again another indication that the ML approach just learns the stark differences between bnab and control data but not what a hiv-1 bnab is). How many sequences are needed to achieve this accuracy (e.g., if you trained only with half of the catnap data, what would the accuracy be?)?

- What's the sequence similarity within the catnap data?

Is there a correlation between the number of examples in the training and the accuracy on the test? So, for example, if there are only a few mper bnab in the database, is prediction accuracy on mper when in the test the worst?

Reviewer #2:

Remarks to the Author:

Foglierini et al. present a machine learning-based approach for the identification of HIV-1 broadly neutralizing antibodies from immune repertoire analyses. The authors use a combination of machine learning techniques trained on data from the CATNAP database to predict whether a particular antibody sequence, converted to relevant features, is a bnAb of a specific set of bnAb classes. The algorithm is used to identify and screen potential bnAbs experimentally in a set of BCR sequences isolated from HIV-1 infected individuals. A single VRC01 class CD4 binding site bnAb was identified, and its binding mode was determined using cryo-EM. The paper is well written and the results are likely to interest a wide audience. My comments/concerns are as follows:

The test set validation description needs additional information. How similar is the training set to the test set? If the training sets contain entries with features that are highly similar to those of the test set, the accuracy of the model could be artificially high. It would be helpful to understand the feature distributions in the training vs. the test.

The experimental validation is interesting but does not support the model's reliability for non-

VRC01 class CD4 binding site antibodies. Several non-CD4 binding site antibodies were identified in the predictions but were not made. Further, of the three CD4 binding site antibodies tested, only one showed breadth. This suggests the model may struggle when identifying novel bnAbs. Could sequence alignment alone have identified these antibodies? The authors should provide sequence identity/similarity distributions to VRC01 in the dataset to compare.

The appearance of a single Fab bound trimer population mixed with a large number of unbound trimer particles in the cryo-EM results is unusual for a CD4 binding site bnAb at six-fold molar excess Fab relative to the trimer. This is especially unusual, considering the measured apparent affinity is 0.4 nM. Avidity could mask a faster off rate. The authors should measure Fab binding affinities to determine whether an off rate sufficient to lead to a single protomer bound trimer in the cryo-EM is reasonable. If not, an alternative explanation is needed.

The clash score and percentage of poor rotamers in the cryo-EM model are quite high. Additional refinement is recommended.

Measurement and, ideally, replicate measure errors should be reported for the BLI affinity reports. Model fits should be shown in a supplemental figure.

Lines 346-347: This sentence appears to be incomplete.

Line 380: "have" should be "has"

Line 384: Should P43 be F43 (Phe43)?

Reviewer #3:

Remarks to the Author:

The manuscript is interesting considering that it arrives at AI based identification of HIV-1 specific broadly neutralizing antibodies and experimentally confirms binding with reasonable affinity to CD4-binding site of the envelop glycoprotein of HIV-1, as well as it shows neutralization capability in case of wide range of clades tested. Further, cryo-EM structure of the Fab fragment of one of the antibodies to CD4 binding domain trimer of the GP120 at 3.7A resolution is being reported. The work has been carried out competently. It is good to observe that atleast in case of (bNAb4251) antibody a very high affinity binding was achieved and it was also the same antibody which showed 80 % of the tested virus clades neutralized. Also, it is the same high affinity antibody for which the Cryo-EM structure has been determined. My only concern is that considering the relatively low resolution of the structure, some interpretations at atomic resolutions appear overinterpreted. At 3.7A resolution, one can discuss about protein surface regions where the two molecules interact. But to describe specific hydrogens bonds, require resolution better than 3 A. Overall, the language also needs to be improved in the tentire manuscript.

Fogliolini et al.: “RAIN: a Machine Learning-based identification for HIV-1 bNABs”. We would like to thank the reviewers for their time reading the manuscript and for providing helpful comments and suggestions. In the revised manuscript, changes in the text appear in green. We provide point-by-point responses below, where the reviewers' comments are in italics followed by our responses.

Responses to Reviewers

Please note that the Reviewer comments are in black, quoted exactly as provided, and our responses are in blue.

Reviewer #1 (Remarks to the Author)

In this article, the authors developed a machine-learning approach for HIV-1 bnab prediction. Briefly, the authors used a dataset of HIV-1 bnabs as well as healthy control data to delineate a classifier for HIV-1 bnab prediction. While this study is of interest and the question is certainly interesting, I have a several comments regarding the approach and the claims made.

We thank the reviewer for finding our study of interest.

Major comments

- The authors claim that their ML approach predicts HIV-1 bnab. I am not so sure about that. As shown by the authors, the bnab seqs are very different from normal antibodies in terms of SHM amount, for example. For me to believe the authors, they would have to show that their algorithm is not only specific versus healthy controls but also versus other infections. For example, if the authors applied their approach to influenza data, their approach should not find any HIV-1 bnabs, although there might be influenza bnabs in the data. In other words, I don't think their approach has learned what a HIV-1 bnab is. Their approach has probably only learned what a very strange looking antibody sequence vs normal control antibodies looks like. Can the authors comment on that?

Response: We thank the Reviewer for his/her suggestion. First, it is important to mention that the variables we define are specific to HIV bNABs, as not all neutralizing antibodies share similar characteristics. To demonstrate the specificity of our approach, we tested our pipeline on Influenza specific repertoires. We used scBCR data obtained from Influenza vaccinated donor at days 7 and 9 post vaccination (Horns et al., 2020). We processed three sequencing runs containing 4'691, 8'222 and 8'052 paired BCRs sequences, respectively. It's worth noting that

these repertoires contain Influenza bNAbs belonging to the lineage clone L3. However, our models did not detect them. Furthermore, no HIV bNAbs was detected by the shared algorithms. These additional data are now present in new **Supplementary Figure 7**.

- I don't understand why the authors need to use 4 different classifiers? what do they each add? and what features are the most important for classifying bnab vs non-bnab?

Response: We decided to use 4 different classifiers to increase robustness and increase confidence in the derived predictions. Indeed, the aim of RAIN is to limit the experimental work of the user. To emphasize it, we have added the following sentence at line 237 of the revised manuscript: “we decided to use different machine learning approaches to increase robustness and decrease the likelihood of false predictions”.

Concerning the choice of the algorithms, the following statements are now present in the revised version of the manuscript.

Line 245: “An anomaly detection (AD) algorithm has been used in the specific case of a binary classification task, where one group appears as an outlier (Steinwart et al., 2005). Given the scarcity of reported HIV-1 bNAbs compared to the quantity of mAb, we first opted for the AD algorithm to automatically identify bNAbs”.

We next mention that a category of bNAbs is not clearly identified by the AD approach, and the precision scores were very low. Therefore, we prompt the usage of other models.

Line 255-258: “However, bNAbs targeting the V3 glycan were poorly identified, with an AUC of 0.64. Moreover, a high number of false positives was obtained, indicating a low precision with the AD (**Figure 4a**). To increase recall and precision of our detection method, we used both Decision Tree (DT) and random forest (RF) algorithms.”

Decision Trees have the advantage to be intuitive and easy to interpret, while Random forests offer high accuracy and robustness by combining multiple decision trees. Random forests have the advantage of scoring features by their importance, thus highlighting the differences between the various antigenic sites targeted by the bNAbs. Finally, Super Learner combines multiple base learners by automatically selecting the best combination of models, making it highly adaptable and effective across various datasets and tasks.

The most important features for classifying HIV bNAbs are presented in **Figure 4e** and further discussed in lines 288-297.

- how similar are train and test data sequence-wise?

Response: To answer this question, it is important to mention that training and test datasets should ideally represent the same underlying population. Still, they are not required to be similar in terms of specific instances or characteristics. In fact, test dataset diversity can enhance the model's robustness and generalizability across scenarios. Nevertheless, to address the sequence-wise similarity between the two sets of data, we generated similarity matrices for the complete VH and the CDRH3 only (**Figure R1**). Interestingly, for the complete VH sequences the highest variability between training and test sets is found for the anti-V1V2 and CD4bs bNAbs. While higher level of similarity can be observed for the others other antigenic sites. As expected, when the CDRH3 sequences are compared, high diversity can be observed and thus for all antigenic sites (**Figure R1**).

Figure R1. Sequence similarity matrices for training and test datasets. Similarity matrices of training versus test sequences with entire VH (left) and CDRH3 only (right). In the heatmaps, sequences are ranked based on their V and J genes. In both cases, matrices were created using ANARCI, and the similarity scores ranging from 0 to 1 indicate the degree of similarity between sequences, with higher scores representing lower Levenshtein distances. The mAbs sequences were down sampled to 100 for the training set and 50 for the test set. Datasets used for each

antigenic site model are represented in the different panels: CD4bs (a), MPER (b), V1V2 apex (c), V3 glycan (d) and Interface (e).

- How similar are bnab and control data sequence-wise?

Response: We thank the Reviewer for this question, and to address it, we generated additional similarity matrices, now present in the manuscript as new Figure (**Figure 2**). We observed some level of similarity between bNAbs and mAbs. The similarity is driven by the framework region (**Figure R2b**). This result was expected since immunoglobulin framework derives from the recombination of a define gene numbers. Moreover, we observed that the highest variance can be attributed to the length of the CDRH3 between anti-V1V2 bNAbs and mAbs.

Figure R2 (Fig. 2 in the modified version of the manuscript). **Sequence similarity matrices of HIV-1 bNAbs and control mAbs.** (a) Similarity matrices for 255 bNAbs grouped by antigenic site for the entire VH (left) or the CDRH3 only (right). (b) Similarity matrices of bNAbs versus mAbs with entire VH (left) and CDRH3 only (right). In the heatmaps, sequences are ranked based on their V and J genes. In both cases, matrices were created using ANARCI, and the similarity scores ranging from 0 to 1 indicate the degree of similarity between sequences, with higher scores representing lower Levenshtein distances. For panel b, mAbs sequences were down sampled to 500 to enable display.

- The catnap dataset is quite small, yet the accuracy is quite high (which is again another indication that the ML approach just learns the stark differences between bnab and control data but not what a hiv-1 bnab is).

Response: We respectfully disagree with the Reviewer and attribute the high accuracy to the class imbalance between bNABs and mAbs (Thölke et al., 2023). To demonstrate it, we measured together with the accuracy, the AUC, the precision, and the recall scores to gain a more comprehensive understanding of the performance across our various approaches (**Figure 4a** and **Supplementary Figure 6**).

How many sequences are needed to achieve this accuracy (e.g., if you trained only with half of the catnap data, what would the accuracy be?)?

Response: All models were trained with 60% of the CATNAP sequences retrieved. This corresponds to the following number of bNABs: 15 MPER, 18 Interface, 26 CD4bs, 34 V3 and 64 V1V2. It is also interesting to note that there is no direct correlation between the number of bNABs in the training set and AUC or precision scores (**Figure R3**). Of note, anomaly detection is not represented in the plot because there are no bNABs in the training dataset; only non-anomaly data (i.e. mAbs) are used to train the model.

Figure R3. Impact of bNABs number on AUC and precision scores. The linear regression plot illustrates the relationship between the number of bNABs in the training dataset and the AUC score (a) or the precision score (b).

- What's the sequence similarity within the catnap data?

Response: To answer this point, we generated an additional similarity matrix shown in **Figure R2a** (also added to the revised manuscript). The similarity matrix reveals low level of similarity for the full VH and CDRH3, only the anti-V1V2 share homology.

Is there a correlation between the number of examples in the training and the accuracy on the test? So, for example, if there are only a few mper bnab in the database, is prediction accuracy on mper when in the test the worst?

Response: As shown in **Figure R3** there is no clear relationship between the precision score or the AUC score and the number of bNAbs in the training dataset.

Reviewer #2 (Remarks to the Author)

Fogliolini et al. present a machine learning-based approach for the identification of HIV-1 broadly neutralizing antibodies from immune repertoire analyses. The authors use a combination of machine learning techniques trained on data from the CATNAP database to predict whether a particular antibody sequence, converted to relevant features, is a bnAb of a specific set of bnAb classes. The algorithm is used to identify and screen potential bnAbs experimentally in a set of BCR sequences isolated from HIV-1 infected individuals. A single VRC01 class CD4 binding site bnAb was identified, and its binding mode was determined using cryo-EM. The paper is well written and the results are likely to interest a wide audience. My comments/concerns are as follows:

We thank the Reviewer for the positive evaluation of our manuscript.

The test set validation description needs additional information.

Response: We added additional information regarding the test set validation lines 275-281, and we have included a new **Supplementary Table 3**, which provides descriptions and performance scores (AUC, accuracy, recall, and precision) for the AD, DT, and RF algorithms on the validation set. Of note, the SL algorithm was trained using 10-fold cross-validation.

Supplementary table 3. Performance metrics of the three algorithms using the validation dataset.

Algo	Ag site	bNAbs number	TP	FP	TN	FN	AUC	Accuracy	Recall	Precision
AD	CD4bs	15	12	22	2966	3	0.90	0.99	0.80	0.35
DT	CD4bs	15	7	4	2984	8	0.73	1.00	0.47	0.64
RF	CD4bs	15	7	0	2988	8	0.90	1.00	0.47	1.00
AD	MPER	3	1	38	2956	2	0.66	0.99	0.33	0.03
DT	MPER	3	2	4	2990	1	0.83	1.00	0.67	0.33
RF	MPER	3	2	0	2994	1	0.83	1.00	0.67	1.00
AD	V1V2 apex	25	19	55	2932	6	0.87	0.98	0.76	0.26
DT	V1V2 apex	25	23	12	2975	2	0.96	1.00	0.92	0.66
RF	V1V2 apex	25	19	0	2987	6	1.00	1.00	0.76	1.00
AD	V3 glycan	9	5	57	2938	4	0.77	0.98	0.56	0.08
DT	V3 glycan	9	6	8	2987	3	0.83	1.00	0.67	0.43
RF	V3 glycan	9	2	0	2995	7	1.00	1.00	0.22	1.00
AD	Interface	3	1	31	2964	2	0.66	0.99	0.33	0.03
DT	Interface	3	3	4	2991	0	1.00	1.00	1.00	0.43
RF	Interface	3	2	0	2995	1	1.00	1.00	0.67	1.00

How similar is the training set to the test set?

Response: We provided the same response to the third question of Reviewer 1; we simply copied our previous answer. We agree that training and test datasets should ideally represent the same underlying population. Still, they are not required to be similar in terms of specific instances or characteristics. In fact, test dataset diversity can enhance the model's robustness and generalizability across scenarios. To address the sequence-wise similarity between the two sets of data, we generated similarity matrices for the complete VH and the CDRH3 only (**Figure R1**). Interestingly, for the complete VH sequences the highest variability between training and test sets is found for the anti-V1V2 and CD4bs bNAbs. While higher level of similarity can be observed for the others other antigenic sites. As expected, when the CDRH3 sequences are compared, high diversity can be observed and thus for all antigenic sites (**Figure R1**).

If the training sets contain entries with features that are highly similar to those of the test set, the accuracy of the model could be artificially high. It would be helpful to understand the feature distributions in the training vs. the test.

Response: This is an important point, the distribution of features may not always be identical between the training and test sets, as illustrated in the **Figure R4 (a-e)**. We also assessed the AUC score of twenty different data split (**Figure R4 f**) using random seed numbers. Due to the small sample size of bNAbs, we observe some variability in the AUC performance evaluation. However, the median remains above 0.75 for the majority of the models. We have highlighted in red the split used in the publication. Even though we did not have the split with the highest AUC performance for the CD4bs, we were able to identify three anti-CD4bs bNAbs.

Figure R4. Distribution of features within the bNAbs training and test datasets. (a-e) Comparison of features distribution of training and test set and (f) assessment of AUC using twenty different data splits. Each point represents a different split, red points represent the split used in the publication.

The experimental validation is interesting but does not support the model's reliability for non-VRC01 class CD4 binding site antibodies. Several non-CD4 binding site antibodies were identified in the predictions but were not made. Further, of the three CD4 binding site antibodies tested, only one showed breadth. This suggests the model may struggle when identifying novel

bnAbs. Could sequence alignment alone have identified these antibodies? The authors should provide sequence identity/similarity distributions to VRC01 in the dataset to compare.

See figure below.

Response: We generated all the antibodies identified by the 4 algorithms. To demonstrate that these antibodies could not have been identified by sequence alignment, we plotted the sequence similarity of the heavy chain to VRC01 **Figure R5**. Among all sequencing runs (D1, D2, B3, G3, G4, S4) analyzed only 17 sequences have more than 65% (maximum = 67 %) sequence identity with VRC01. Furthermore, none of CD4bs bNAbs identified by RAIN are among those 17 sequences.

Figure R5. Sequence similarity distributions to VRC01. Plotted is the percentage of similarity to VRC01 with each antibody sequence represented as a point. The dashed line in G4 represents the similarity of bNAb4251 or bNAb2101 with VRC01 (57% of similarity). The dashed line in S4 represents the similarity of bNAb1586 with VRC01 (54%).

In conclusion, it is very unlikely that sequence similarity or homology could have been used to identify the bNAbs. This has also been emphasized in the text lines 174-175. “This result indicates that a homology and alignment approach to identify bNAbs would probably be unsuccessful. “

The appearance of a single Fab bound trimer population mixed with a large number of unbound

trimer particles in the cryo-EM results is unusual for a CD4 binding site bnAb at six-fold molar excess Fab relative to the trimer. This is especially unusual, considering the measured apparent affinity is 0.4 nM. Avidity could mask a faster off rate. The authors should measure Fab binding affinities to determine whether an off rate sufficient to lead to a single protomer bound trimer in the cryo-EM is reasonable. If not, an alternative explanation is needed.

Response: First, we thank the reviewer for pointing out a typographical mistake, Fab was added at 3-fold molar excess. Nevertheless, we believe that we obtained a single Fab4251 per trimer due to the short incubation time (10 minutes) between a three-fold molar excess of bNAb4251 and SOSIP. We also performed BLI measurement with Fabs and determine a K_D of 5 ± 2.4 nM, and 17.5 ± 4 nM for Fab4251 and Fab2101, respectively (**Supplementary Figure 9**). To further demonstrate the interaction, we performed negative staining with the Fabs in complex with SOSIP trimer and obtained 2 Fab4251/Trimer and 1 Fab2101/Trimer (new **Figure 6a-b** of the manuscript). These aspects are specified in the text. Line 369: “To further characterize these interactions, we calculated the affinity of the fragment antigen binding (Fab) to SOSIP trimers and obtained the following K_D 5 ± 2.4 nM, and 17.5 ± 4 nM for Fab2101 and Fab4251 respectively (**Supplementary figure 9b**). Of note, Fab1586 demonstrated poor affinity with a K_D measure of 1μ M (**Supplementary figure 9b**).”

Figure R6. Fab4251 and Fab2101 interaction with BG505 DS-SOSIP. (a) 3D reconstruction of Fab4251-SOSIP complex by nsEM. **(b)** 3D reconstruction of Fab2101-SOSIP complex by nsEM.

The clash score and percentage of poor rotamers in the cryo-EM model are quite high. Additional refinement is recommended.

Response: After additional refinement on the structure our model has now a percentage of poor rotamers of 2.3 and a clash score of 9.0 which we believe is acceptable at 3.8 \AA resolution for SOSIP. As example here are some SOSIP cryoEM structure with clash score in the same range: PDB 7TFN clash score: 14.1 (Yang et al., 2022), PDB 7TFO clash score: 16.4 (Yang et al.,

2022), PDB 5V8M clash score: 5.88 (Molinos-Albert et al., 2023), PDB 7LOK clash score: 14.4 (Jette et al., 2021). Taking in account the resolution obtained we removed some of our statements concerning side chains interaction and hydrogen bonds.

Measurement and, ideally, replicate measure errors should be reported for the BLI affinity reports. Model fits should be shown in a supplemental figure.

Response: We thank the Reviewer for pointing out this omission. We added the standard deviation of our different experiments. We also performed measurement with Fabs and curves, fitting are shown for Ab and Fabs in the **Supplementary Figure 9**.

Lines 346-347: This sentence appears to be incomplete.

Line 380: “have” should be “has”

Line 384: Should P43 be F43 (Phe43)?

Response: Thank you those mistakes have been corrected.

Reviewer #3 (Remarks to the Author):

The manuscript is interesting considering that it arrives at AI based identification of HIV-1 specific broadly neutralizing antibodies and experimentally confirms binding with reasonably affinity to CD4-binding site of the envelop glycoprotein of HIV-1, as well as it shows neutralization capability in case of wide range of clades tested. Further, cryo-EM structure of the Fab fragment of one of the antibodies to CD4 binding domain trimer of the GP120 at 3.7A resolution is being reported. The work has been carried out competently. It is good to observe that atleast in case of (bNAb4251) antibody a very high affinity binding was achieved, and it was also the same antibody which showed 80 % of the tested virus clades neutralized. Also, it is the same high affinity antibody for which the Cryo-EM structure has been determined.

We thank the reviewer for his/her interest on our manuscript.

My only concern is that considering the relatively low resolution of the structure, some interpretations at atomic resolutions appear overinterpreted. At 3.7A resolution, one can

discuss about protein surface regions where the two molecules interact. But to describe specific hydrogens bonds, require resolution better than 3 Å.

Response: We agree, and we removed some of our statements on hydrogen bond and lateral chain placement.

Overall, the language also needs to be improved in the tentative manuscript.

Response: The manuscript has been proofread and corrected by a native English speaker.

References

- Horns, F., Dekker, C.L., and Quake, S.R. (2020). Memory B Cell Activation, Broad Anti-influenza Antibodies, and Bystander Activation Revealed by Single-Cell Transcriptomics. *Cell reports* 30, 905-913.e906. <https://doi.org/10.1016/j.celrep.2019.12.063>.
- Jette, C.A., Barnes, C.O., Kirk, S.M., Melillo, B., Smith, A.B., and Bjorkman, P.J. (2021). Cryo-EM structures of HIV-1 trimer bound to CD4-mimetics BNM-III-170 and M48U1 adopt a CD4-bound open conformation. *Nature communications* 12, 1950. 10.1038/s41467-021-21816-x.
- Molinos-Albert, L.M., Baquero, E., Bouvin-Pley, M., Lorin, V., Charre, C., Planchais, C., Dimitrov, J.D., Monceaux, V., Vos, M., Hocqueloux, L., et al. (2023). Anti-V1/V3-glycan broadly HIV-1 neutralizing antibodies in a post-treatment controller. *Cell host & microbe* 31, 1275-1287.e1278. <https://doi.org/10.1016/j.chom.2023.06.006>.
- Steinwart, I., Hush, D., and Scovel, C. (2005). A Classification Framework for Anomaly Detection. *J. Mach. Learn. Res.* 6, 211–232.
- Thölke, P., Mantilla-Ramos, Y.-J., Abdelhedi, H., Maschke, C., Dehgan, A., Harel, Y., Kemtur, A., Mekki Berrada, L., Sahraoui, M., Young, T., et al. (2023). Class imbalance should not throw you off balance: Choosing the right classifiers and performance metrics for brain decoding with imbalanced data. *NeuroImage* 277, 120253. <https://doi.org/10.1016/j.neuroimage.2023.120253>.
- Yang, Z., Dam, K.A., Bridges, M.D., Hoffmann, M.A.G., DeLaitsch, A.T., Gristick, H.B., Escolano, A., Gautam, R., Martin, M.A., Nussenzweig, M.C., et al. (2022). Neutralizing antibodies induced in immunized macaques recognize the CD4-binding site on an occluded-open HIV-1 envelope trimer. *Nature communications* 13, 732. 10.1038/s41467-022-28424-3.

Reviewers' Comments:

Reviewer #1:

Remarks to the Author:

The authors have addressed all of my comments.

Reviewer #2:

Remarks to the Author:

All of my concerns were adequately addressed.

Foglierini et al.: “RAIN: Machine Learning-based identification for HIV-1 bNAbs”. We would like to thank the reviewers for their time reading the manuscript and for providing helpful comments and suggestions.

Responses to Reviewers

Reviewer #1 (Remarks to the Author)

The authors have addressed all of my comments.

We thank the reviewer for his/her helpful question and suggestion during the reviewing process. He or She has greatly helped to improve the quality of the manuscript.

Reviewer #2 (Remarks to the Author)

All of my concerns were adequately addressed.

We thank the reviewer for his/her helpful question and suggestion during the reviewing process. He or She has greatly helped to improve the quality of the manuscript.